# *k*-Circle Formation and *k*-epf by Asynchronous Robots

**Subhash Bhagat [1], Bibhuti Das [2], Abhinav Chakraborty [2] and Krishnendu Mukhopadhyaya [2,\*]**

[1] Indian Association for the Cultivation of Science, Kolkata 700108, India; subhash.bhagat@niser.ac.in or subhash.bhagat.math@gmail.com

[2] Advanced Computing and Microelectronics Unit, Indian Statistical Institute, Kolkata 700108, India; bibhuti_r@isical.ac.in or dasbibhuti905@gmail.com (B.D.); abhinav_r@isical.ac.in or abhinav.chakraborty06@gmail.com (A.C.)

\* Correspondence: krishnendu@isical.ac.in or krishnendu.mukhopadhyaya@gmail.com

**Abstract:** For a given positive integer $k$, the $k$-circle formation problem asks a set of autonomous, asynchronous robots to form disjoint circles having $k$ robots each at distinct locations, centered at a set of fixed points in the Euclidean plane. The robots are identical, anonymous, oblivious, and they operate in Look–Compute–Move cycles. This paper studies the $k$-circle formation problem and its relationship with the $k$-epf problem, a generalized version of the embedded pattern formation problem, which asks exactly $k$ robots to reach and remain at each fixed point. First, the $k$-circle formation problem is studied in a setting where the robots have an agreement on the common direction and orientation of one of the axes. We have characterized all the configurations and the values of $k$, for which the $k$-circle formation problem is deterministically unsolvable in this setting. For the remaining configurations and the values of $k$, a deterministic distributed algorithm has been proposed, in order to solve the problem. It has been proved that for the initial configurations with distinct robot positions, if the $k$-circle formation problem is deterministically solvable then the $k$-epf problem is also deterministically solvable. It has been shown that by modifying the proposed algorithm, the $k$-epf problem can be solved deterministically.

**Keywords:** swarm robotics; $k$-circle formation; $k$-epf; asynchronous; one axis agreement





## 1. Introduction

A swarm of robots is a multi-robot system consisting of small and inexpensive mobile robots working together in a cooperative environment to achieve some specific goal. A swarm of robots has the potential to be utilized in risky and hazardous scenarios, such as in the fields of search and rescue operations, military operations, fire fighting, etc. Robots are autonomous, anonymous, homogeneous, and oblivious. A robot is modeled as a geometric point in the plane. The robots do not have any agreement on a global coordinate system. Each robot has its own local coordinate system. They do not have any explicit means of communication. When a robot becomes active, it operates in Look–Compute–Move (LCM) cycles. The study of such a distributed system of mobile robots aims at finding a minimal set of capabilities in order to solve a given problem. Some of the well-known co-ordination problems studied in this area of research work are gathering and convergence [1–9], arbitrary pattern formation [10–13], embedded pattern formation [14,15], circle formation [16–21] etc.

The embedded pattern formation [14,15] asks the robots to form a given pattern, in which the points comprising the pattern are assumed to be pre-fixed and visible to all the robots like landmarks. Cicerone et al. [22] studied the gathering on meeting points problem where the robots need to gather at one of the pre-determined points in the plane, referred to as meeting points. In the discrete domain, Bhagat et al. [23] studied the gathering over meeting nodes problem in an infinite grid, where some of the nodes of the grid have been considered as meeting nodes.

In this paper, we have considered $m > 0$ fixed points in the Euclidean plane. For a given positive integer $k$, we consider $n$ mobile robots such that $n = km$. The $k$-circle

formation problem is defined as follows: the *n* mobile robots need to form *m* circles, each centered at one of the fixed points, such that each circle contains exactly *k* robots at distinct positions. The circles formed by the robots must be disjoint, i.e., no robot is allowed to lie on any two distinct circles. Note that the circles do not require to be necessarily uniform. In other words the *k* robots forming one particular circle have identical distance from a particular fixed point, but their mutual distances can be anything. In general, one may consider different radii for the circles. In this paper, we consider all the circles to have the same radius, which is a special case of the *k*-circle formation problem.

We have considered a swarm of mobile robots, which are represented as points in the Euclidean plane. The robots are silent, i.e., they do not have any explicit direct way of communication. They coordinate their movements by perceiving the positions of other robots. The robots are assumed to have unlimited visibility, i.e., the robots can observe the entire plane. We assume that initially all the robots are placed at distinct locations in the plane. Robots are assumed to be:

- Autonomous, i.e., they do not have any centralized controller;
- Anonymous, i.e., they have no unique identifier;
- Oblivious, i.e., they do not remember anything about the past events;
- Homogeneous, i.e., they execute the same deterministic algorithm.

The robots operate in Look–Compute–Move (LCM) cycles. In the look phase, the robot senses the current configuration in its own local coordinate system. In the compute phase, a destination point is computed by the robot. The destination point may be the robot's current location. In the move phase, the robot moves towards the destination point. A robot cannot distinguish between a static robot and a moving robot, in its look phase.

The robots have non-rigid motion, i.e., there exists a fixed but unknown $\delta > 0$ such that if the destination is more than $\delta$ distance away from a robot, the robot moves at least $\delta$ amount towards its destination. If the destination point is less than $\delta$ distance away, the robot is guaranteed to reach its destination.

The scheduler is assumed to be fair, i.e., each robot is activated infinitely often and performs its LCM cycle within finite time. The following types of schedulers are commonly used:

1. Fully-synchronous (FSYNC): Robots have a common notion of time. All the robots are activated simultaneously and perform all operations synchronously.
2. Semi-synchronous (SSYNC): It coincides with the FSYNC scheduler with the only difference that not all the robots are activated in each round.
3. Asynchronous (ASYNC): Robots do not have a common notion of time. They are activated independently, and the duration of each LCM cycle and inactivity phase is finite but unbounded.

We have considered a fair ASYNC scheduler. Due to the asynchrony, a robot might be in motion when some other robot is in its look phase. As a result, for an active robot, the perceived configuration in its look phase might be different from the configuration at the time when it actually starts moving.

The theoretical motivation for studying the *k*-circle formation problem is twofold. First, we believe that the problem is theoretically interesting as it is a hybrid problem in between the partitioning problem [24] and the circle formation problem [16–21]. Both the problems individually differs from the *k*-circle formation problem w.r.t. the following points:

1. The partitioning problem asks the robots to divide themselves into *m* groups, each having *k* robots. In addition, the robots in each group are asked to converge into a small area. Unlike the *k*-circle formation problem, the robots do not need to form circles containing exactly *k* robots, centered at one of the pre-fixed points.
2. The circle formation problem asks the robots to place themselves at distinct locations on a circle (not defined a priori), within finite amount of time. In this problem, all the robots participate in forming one single circle, whereas, in the *k*-circle formation

problem, the robots need to form $m$ circles each containing exactly $k$ robots and centered at one of the fixed points.

To the best of our knowledge, we believe that this is the first work that aims at connecting the two well-known problems in the literature, namely the partitioning problem and the circle formation problem. Secondly, if the robots could solve the *k-circle formation* problem, then all the $k$ robots which lie on the same circle can gather at their respective center, which is a fixed point, within finite number of moves. Thus, studying the solvability of the $k$-circle formation problem includes investigating the solvability of a generalized version of the embedded pattern formation problem (referred to as *k*-epf problem in this paper), where $k$ robots need to reach and remain at each fixed point.

In addition, we believe that the $k$-circle formation problem would have the following applications in the field of swarm robotics:

1.  The set of fixed points can be considered as emergency points, which need to be surrounded. By solving the $k$-circle formation problem, a swarm of robots can divide themselves into groups, containing $k$ robots and build a perimeter, surrounding the emergency points.
2.  The set of fixed points can also be considered as charging stations, with some given permitted capacity. The robots need to be charged after a certain amount of time to continue working. By solving $k$-circle formation problem, the robots can reach the charging stations without violating the permitted capacity.

### 1.1. Related Works

The arbitrary pattern formation problem asks the robots to form an arbitrary geometric pattern given as an input. Suzuki et al. [10] studied the formation of geometric patterns in the plane by the robots. They characterized the class of geometric patterns that can be formed by the robots in terms of their initial configurations. Flocchini et al. [11] investigated the solvability of the arbitrary pattern formation problem by asynchronous robots. Fujinaga et al. [12] proposed an algorithm that forms a given pattern $P$ starting from any initial configuration $I$, if $\rho(I)$ divides $\rho(P)$, where $\rho(.)$ denotes the geometric symmetricity, provided that both $I$ and $P$ do not contain multiplicities. Cicerone et al. [13] studied the arbitrary pattern formation problem without assuming common chirality. Bose et al. [25] investigated the arbitrary pattern formation problem for asynchronous robots with lights. The presence of lights serves both as a medium of weak explicit communication and also as a form of memory. This work investigates the problem in a setting where the view of a robot is obstructed by the presence of other robots. In the discrete domain, Bose et al. [26] investigated the arbitrary pattern formation problem on infinite grid for asynchronous robots.

Defago et al. [27] studied the circle formation problem for oblivious anonymous mobile robots without chirality. Defago et al. [18] proposed a distributed algorithm, that ensures that the robots deterministically form a non-uniform circle in a finite number of steps and converge towards a solution to the uniform circle formation. The circle formation problem for asynchronous fat robots has been studied in [17,19]. Flocchini et al. [20] proved that the uniform circle formation problem is solvable for any initial configuration in which the robots are in distinct locations. Bhagat et al. [16] proposed a distributed algorithm to solve the circle formation problem by asynchronous robots having one bit of persistent memory, which minimizes the maximum distance traveled by any robot. Mondal et al. [21] proposed a distributed algorithm for fat robots having limited visibility to form a uniform circle. Felleti et al. [28] studied the uniform circle formation for opaque robots with lights.

The landmarks covering problem studied by Fujinaga et al. [14], asks the robots to reach a configuration where at each landmark point (which is a pre-fixed point in the plane), there is precisely one robot. They proposed an algorithm, assuming common chirality among the robots, which minimizes the total distance traveled by all the robots. Cicerone et al. [15] studied the embedded pattern formation problem without assuming common chirality. Cicerone et al. in [22] solved a variant of the gathering problem,

where the robots need to gather at one of the pre-determined points, referred to as meeting points. They also studied this problem w.r.t. two objective functions, by minimizing the total distance traveled by all robots and by minimizing the maximum distance traveled by a single robot. Bhagat et al. [23] studied the gathering over meeting nodes problem in an infinite grid.

Efrima et al. studied the effect of the common orientation on the solvability of the partitioning problem [24] for homogeneous robots. Z. Liu et al. [29] studied the team assembling problem by heterogeneous robots. Given robots with $k$ different colors, the robots are required to partition themselves into teams satisfying a given specification $A = (a_1, a_2, \ldots, a_k)$, where $a_i$ is the number of robots with color $i$ in one team.

### 1.2. Our Contributions

This paper studies the $k$-circle formation problem for a set of asynchronous oblivious mobile robots in the Euclidean plane. This problem is a hybrid between the partitioning problem and the circle formation problem. The relationship between the $k$-circle formation problem and the $k$-epf problem is investigated. The $k$-circle formation problem has been studied here in a setting where the robots have an agreement on the direction and orientation of one axis. This paper contributes the following three results:

1.　All the initial configurations and the values of $k$ for which the $k$-circle formation problem is deterministically unsolvable have been characterized.
2.　For the remaining initial configurations and the values of $k$, a deterministic distributed algorithm has been proposed to solve the problem under an asynchronous scheduler.
3.　For the initial configurations with distinct robot positions, if the $k$-circle formation problem is deterministically solvable then the $k$-epf problem is also deterministically solvable.

### 1.3. Outline

In the next section, we formally define the model and introduce all the required notations and definitions. Section 3 discusses the impossibility result for the $k$-circle formation problem. Section 4 describes the proposed algorithm that deterministically solves the $k$-circle formation problem. Section 5 discusses the correctness of the proposed algorithm. Section 6 discusses the relationship between the $k$-circle formation problem and the $k$-epf problem. Finally, Section 7 concludes the paper.

## 2. Model and Definitions

The robots are assumed to be dimensionless, oblivious, anonymous, autonomous, and homogeneous. They are represented by points in the Euclidean plane. They have unlimited visibility range and have no explicit way of communication. The movements of robots are non-rigid. The state of a robot can be either active or inactive. They execute Look–Compute–Move (LCM) cycle when they become active. We have considered a fair ASYNC scheduler, i.e., each robot is activated infinitely often and the duration of each LCM cycle is finite but unbounded. The robots have one axis agreement, i.e., they agree on the direction and orientation of any one of the axes. We assume that they have an agreement on the $y$-axis. The following notations have been used in the proposed algorithms.

- **Configuration:** Let $R = \{r_1, r_2, \ldots, r_n\}$ be the set of robots. Let $r_i(t)$ denote the position of the robot $r_i$ at time $t$. $R(t) = \{r_1(t), r_2(t), \ldots, r_n(t)\}$ is the set of robot positions at time $t$. We are given a set of fixed points denoted by $F = \{f_1, f_2, \ldots, f_m\}$. It is assumed that $n = km$ for some positive integer $k$. Let $c$ be the center of gravity of the set of fixed points $F$. We assume that the $y$-axis passes through $c$. We also assume that $c$ is the origin. Let $F_y$ and $R_y(t)$ denote the set of fixed points and robot positions, respectively, on the $y$-axis at time $t$. Let $d(r, f)$ denote the Euclidean distance between $r$ and $f$. The pair $C(t) = (R(t), F)$ represents the configuration at time $t$. In an initial configuration $C(0)$, it is assumed that all the robots are stationary and are placed at distinct positions. A configuration is said to be balanced at time $t$ if the number of

robots in both the open half-planes delimited by the $y$-axis is equal. Otherwise, the configuration is said to be unbalanced.

- **Circles and radii of circles:** We consider that all the circles formed by the robots would have the same radius. Let $\rho$ denote the radius of the circles. Furthermore, let $C(f, \rho)$ denote the circle centered at $f \in F$ with radius $\rho$. We have used the following notations to formulate the radius $\rho$ of the circles:

  1. $\rho_1$ = minimum distance between two fixed points.
  2. $\rho_2$ = minimum distance between a fixed point $f \in (F \setminus F_y)$ and the $y$-axis.

  The radius $\rho$ is defined as $\rho = \dfrac{1}{3} \min(\rho_1, \rho_2)$.

- We call a configuration $C(t)$ final if the following conditions hold:

  1. Every robot $r_i$ is on a circle $C(f_j, \rho)$ for some $f_j \in F$,
  2. $C(f_i, \rho) \cap C(f_j, \rho) = \emptyset$ for $f_i \neq f_j$,
  3. Each circle contains exactly $k$ robots at distinct positions.

  The $k$-circle formation problem asks the robots to reach and remain in the final configuration, starting from an initial configuration.

- A fixed point and its respective circle $C(f_j, \rho)$ are said to be unsaturated, if $C(f_j, \rho)$ contains less than $k$ robots on it. Let $N_j(t)$ denote the deficit in the number of robots in order to have exactly $k$ robots on the $C(f_j, \rho)$. A fixed point and its respective circle $C(f_j, \rho)$ are said to be saturated, if $C(f_j, \rho)$ contains exactly $k$ robots on it. In case $C(f_j, \rho)$ contains more than $k$ robots, then $C(f_j, \rho)$ and $f_j$ are called oversaturated.

- **Configuration Rank.** Let $y(s_i)$ denote the $y$-coordinate of a point $s_i$. Note that the robots do not have an agreement on the positive direction of the $x$-axis. In case, the robots could have an agreement on the positive direction of the $x$-axis, $\beta(s_i)$ denotes the $x$-coordinate of $s_i$. Otherwise, $\beta(s_i)$ denotes the distance of $s_i$ from the $y$-axis. The pair $\gamma(s_i) = (\beta(s_i), y(s_i))$ is the configuration rank of the point $s_i$. Between the two points $s_i$ and $s_j$, $s_i$ is said to have higher configuration rank than $s_j$, if $y(s_i) > y(s_j)$ or $y(s_i) = y(s_j)$ and $\beta(s_i) > \beta(s_j)$. Since the robots have unlimited visibility, they can compute the configuration rank of each point $s_i \in F \cup R(t)$.

- **Symmetry about the y-axis.** If the robots $r_i$ and $r_j$ for $i \neq j$, have the same configuration rank, i.e., $\gamma(r_i(t)) = \gamma(r_j(t))$, they are said to be symmetric about the $y$-axis. Let $\phi(r)$ denote the symmetric image of $r$ about the $y$-axis. If robots $r_i$ and $r_j$ are symmetric about the $y$-axis, then $r_i = \phi(r_j)$ and $r_j = \phi(r_i)$. Similarly, two fixed points are said to be symmetric about the $y$-axis, if they have the same configuration rank. An active robot in its look phase identifies the set $R(t)$ to be symmetric about the $y$-axis, if each robot position $r \in R(t)$ has a symmetric image $\phi(r) \in R(t)$. Similarly, a robot can identify whether the set $F$ is symmetric about the $y$-axis or not. An active robot in its look phase identifies the configuration to be symmetric about the $y$-axis if both the sets $F$ and $R(t)$ are symmetric about the $y$-axis. Since the robots have an agreement on the direction and orientation of the $y$-axis, the configuration cannot admit translational symmetry or rotational symmety.

- **Partitioning of configurations:** All the configurations can be partitioned into the following disjoint classes:

  1. $\mathcal{I}_1$– All configurations for which the $y$-axis is not a line of symmetry for $F$ (Figure 1a).
  2. $\mathcal{I}_2$– All configurations for which the $y$-axis is a line of symmetry for $F$, but it is not a line of symmetry for $R(t)$ (Figure 1b).
  3. $\mathcal{I}_3$– All configurations for which the $y$-axis is a line of symmetry for $F \cup R(t)$ and $R_y(t) \neq \emptyset$, i.e., there exists a robot position on the $y$-axis (Figure 1c).
  4. $\mathcal{I}_4$– All configurations for which the $y$-axis is a line of symmetry for $F \cup R(t)$. Furthermore, $F_y = \emptyset$ and $R_y(t) = \emptyset$, i.e., there are no robot positions and fixed points on the $y$-axis (Figure 2a).

5.  $\mathcal{I}_5-$ All configurations for which the $y$-axis is a line of symmetry for $F \cup R(t)$. Furthermore, $F_y \neq \varnothing$ and $R_y(t) = \varnothing$, i.e., there are no robot positions on the $y$-axis, but there are fixed points on the $y$-axis (Figure 2b).

Note that the classification of the configuration depends only on the $y$-axis and $c$. Since the $y$-axis and $c$ are the same for all the robots, they can easily classify a configuration without conflict.

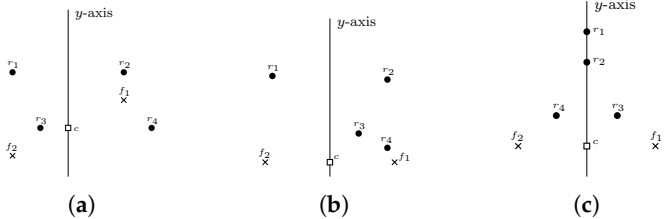

**Figure 1.** Square represents the center of gravity, black circles represent robot positions, and crosses represent fixed points. (**a**) $\mathcal{I}_1$-configuration. (**b**) $\mathcal{I}_2$-configuration. (**c**) $\mathcal{I}_3$-configuration.

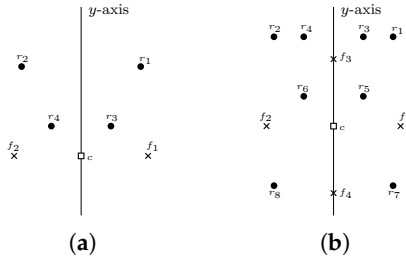

**Figure 2.** (**a**) $\mathcal{I}_4$-configuration. (**b**) $\mathcal{I}_5$-configuration.

## 3. Impossibility Result

In this section, we characterize the initial configurations for which the $k$-circle formation problem cannot be solved deterministically. Notice that if $k$ is an odd integer and the initial configuration $C(0) \in \mathcal{I}_5$, then $|F|$ must be even. For an initial configuration $C(0)$ which is symmetric about the $y$-axis, if both the values of $k$ and $|F|$ are odd, then $R_y(0) \neq \varnothing$. As a result, $C(0)$ cannot possibly belong to $\mathcal{I}_5$.

**Theorem 1.** *If the initial configuration $C(0) \in \mathcal{I}_5$ and $k$ is an odd integer, then the $k$-circle formation problem is deterministically unsolvable.*

**Proof of Theorem 1.** If possible, let algorithm $\mathcal{A}$ solve the $k$-circle formation problem starting from the given initial configuration $C(0) \in \mathcal{I}_5$ when $k$ is odd. Consider the scheduler to be semi-synchronous with the additional property that whenever a robot $r$ is activated, $\phi(r)$ is also activated. We assume that all the robots move with constant speed (which is the same for all robots) without transient stops. We also assume that the distance traveled by $r$ is the same as that by $\phi(r)$. First, consider that both $r$ and $\phi(r)$ have opposite notions of positive $x$-axis direction. As a result, their views would be identical. Since they run the same algorithm, their destinations and the corresponding paths would be mirror images. Even with non-rigid motion, if they travel the same distance, their final positions would be mirror images of each other. Since we started with a symmetric configuration, no algorithm can break the symmetry under this setup. Let $f$ be a fixed point on the $y$-axis. Since the overall configuration is symmetric, the robot positions on $C(f, \rho)$ must be symmetric around the $y$-axis. As $k$ is odd, $C(f, \rho)$ must contain a robot position on the $y$-axis. Since the initial configuration did not have any robot position on the $y$-axis and all the robots move in pairs, having a robot $r$ moved to the $y$-axis would mean moving $\phi(r)$ to the same point. As a result, a point of multiplicity will be created, from which it is

deterministically impossible to separate $r$ and $\phi(r)$. Hence, the $k$-circle formation problem is deterministically unsolvable. $\square$

Notice that the unsolvability criterion (Theorem 1) for the $k$-circle formation problem would never be satisfied when $k$ is an even integer. Even for odd values of $k$ and the symmetric configurations in $\mathcal{I}_3 \cup \mathcal{I}_4$, the unsolvability criterion (Theorem 1) for the $k$-circle formation problem would never be satisfied.

## 4. Algorithm

In this section, we propose a deterministic distributed algorithm that solves the $k$-circle formation problem for the remaining configurations. Each active robot will execute the proposed algorithm $AlgorithmOneAxis(C(t))$ unless $C(t)$ is a final configuration. Each robot will follow the following steps during an execution of $AlgorithmOneAxis(C(t))$:

1. The robots identify the current configuration. The robots agree upon the positive direction of the $x$-axis in some configurations.
2. One or two unsaturated fixed points are selected for the circle formation, referred to as target fixed points.
3. The robots identify one or two robots for each target fixed point, referred to as candidate robots.
4. Each candidate robot moves towards the $k$-circle centered at its target fixed point.

First, we define progress in a half-plane. Let $HL_1$ and $HL_2$ denote the two half-planes delimited by the $y$-axis. Let $f_i$ be the unsaturated fixed point, which has the highest rank in $HL_1$ at time $t \geq 0$. Similarly, suppose $f_j$ is the the unsaturated fixed point, which has the highest rank in $HL_2$ at time $t \geq 0$. We say that there has been more progress in $HL_1$ than $HL_2$ at time $t$ if one of the following conditions holds:

1. $\gamma(f_i) < \gamma(f_j)$;
2. $\gamma(f_i) = \gamma(f_j)$ and $N_i(t) < N_j(t)$;
3. $\gamma(f_i) = \gamma(f_j)$ and $N_i(t) = N_j(t)$ and $d(f_i, r_1(t)) < d(f_j, r_2(t))$ where $r_1$ and $r_2$ are candidate robots for $f_i$ and $f_j$, respectively.

Otherwise, we say that there has been the same progress in both the half-planes.

### 4.1. Agreement One Axis

Since the robots have an agreement on the direction and orientation of the $y$-axis, they also have an agreement on the orientation of the $x$-axis without direction. This is the algorithm by which the robots identify the configurations in which they could have an agreement on the direction of the $x$-axis. The robots make an agreement on the direction of the $x$-axis in such configurations. Each robot in its look phase identifies the class in which $C(t)$ belongs to and considers the following cases:

1. $C(t) \in \mathcal{I}_1$, i.e., $F$ is asymmetric about the $y$-axis. Let $hline_1, \ldots, hline_s$ denote all the horizontal lines, each one of which passes through at least one fixed point, listed according to their increasing $y$-coordinates. Since the fixed points are asymmetric about the $y$-axis, at least one of these lines must contain asymmetric fixed points. Let $hline_v$ be the topmost among such horizontal lines which contains an asymmetric fixed point. Consider the fixed point closest to the $y$-axis and not having a symmetric image on $hline_v$. The direction from the $y$-axis towards the half-plane containing this fixed point is considered as the positive $x$-direction. All the robots agree upon this agreement.
2. $C(t) \in \mathcal{I}_2$, i.e., $F$ is symmetric about the $y$-axis, but $R(t)$ is asymmetric about the $y$-axis. The robots consider the following cases:
   (a) The configuration is unbalanced. The direction from the $y$-axis, towards the half-plane containing the maximum number of robots, is considered as the positive $x$-direction. All the robots agree upon this agreement.

(b) The configuration is balanced and all the fixed points in one of the half-planes are either saturated or oversaturated. In this case the robots consider the positive $x$-direction towards the half-plane in which all the fixed points are either saturated or oversaturated.

(c) The configuration is balanced with at least one unsaturated fixed point in both the half-planes and $R_y(t) \neq \emptyset$. The robots do not make an agreement on the direction of positive $x$-axis. The robots decide to transform the configuration into an unbalanced configuration. Let $r$ be the topmost robot on the $y$-axis. Define $\lambda = \max\limits_{f \in F, \, r_i \in R(t) \setminus \{r\}} d(r_i(t), f)$. Suppose $p$ denotes the point on the $y$-axis, which is at $2\lambda$ distance above from topmost horizontal line $hline_s$. If the position of $r$ is below $p$, then it moves towards $p$ along the $y$-axis. Otherwise, $r$ is moved to one of the half-planes to a point at $\frac{1}{3}\rho$ from the $y$-axis. This upward movement is required to avoid any collision, which might arise due to the inherent motion of $r$ in a half-plane for some $t' \geq t$.

(d) The configuration is balanced with at least one unsaturated fixed point in both the half-planes and $R_y(t) = \emptyset$. The robots consider the following cases:

    i. $k$ is odd and $F_y \neq \emptyset$. Note that in this case, the configuration has an even number of fixed points. The direction from the $y$-axis towards the half-plane in which there has been more progress is considered as the positive $x$-axis direction. It is possible that initially there has been the same progress in both the half-planes. Since $C(0)$ is asymmetric, there must be one asymmetric robot position about the $y$-axis. The positive $x$-direction is considered towards the half-plane that contains the asymmetric robot position, which has the highest configuration rank. All the robots agree upon this agreement.

    ii. Otherwise, the robots do not agree upon the direction of positive $x$-axis direction. This case includes the configurations in which (i) $k$ is even and $F_y \neq \emptyset$, (ii) $k$ is even and $F_y = \emptyset$, and (iii) $k$ is odd and $F_y = \emptyset$. Notice that a configuration in this case might become symmetric with $R_y(t) = \emptyset$. Since the robots are oblivious, they would identify the configuration to be in $\mathcal{I}_4$ or $\mathcal{I}_5$, in which they cannot make an agreement on the direction of positive $x$-axis. This decision of not to agree upon the direction of positive $x$-axis direction would ensure that the robots follow the same strategy in both symmetric and asymmetric cases.

3. $C(t) \in \mathcal{I}_3$, i.e., $F \cup R(t)$ is symmetric about the $y$-axis and $R_y(t) \neq \emptyset$. Since $R(t)$ is symmetric about the $y$-axis, the configuration is balanced. The robots decide to transform the configuration into an unbalanced configuration. The robots follow the same strategy as described in the case of a balanced $\mathcal{I}_2$ configuration with at least one unsaturated fixed point in both the half-planes and $R_y(t) \neq \emptyset$ (case 2(c)).

4. $C(t) \in \mathcal{I}_4$, i.e., $F \cup R(t)$ is symmetric about the $y$-axis, and $F_y = \emptyset$ and $R_y(t) = \emptyset$. Since $R(t)$ is symmetric about the $y$-axis, the configuration is balanced. As there are no robot positions on the $y$-axis, the configuration cannot be transformed into an unbalanced configuration. The robots cannot have an agreement on the direction of positive $x$-axis direction in this case.

5. $C(t) \in \mathcal{I}_5$, i.e., $F \cup R(t)$ is symmetric about the $y$-axis, and $F_y \neq \emptyset$ and $R_y(t) = \emptyset$. In this case, we have a balanced configuration. Since there are no robot positions on the $y$-axis, the configuration cannot be transformed into an unbalanced configuration. Note that $k$ is an even integer in this case. Otherwise, the $k$-circle formation problem is unsolvable. The robots cannot have an agreement on the direction of positive $x$-axis direction in this case.

### 4.2. Target FP Selection

This is the algorithm by which the robots select a target fixed point for the $k$-circle formation. The robots consider the following cases:

1. Robots have an agreement on the positive direction of the $x$-axis. Among the unsaturated fixed points, let $f_j$ be the one, which has the highest configuration rank. The robots select $f_j$ as the target fixed point.

2. Robots do not have an agreement on the positive direction of the $x$-axis. The robots consider the following cases:

    (a) All the fixed points in $F \setminus F_y$ are saturated. Among the unsaturated fixed points in $F_y$, let $f_j$ be the topmost one. The robots select $f_j$ as the target fixed point.

    (b) There exists an unsaturated fixed point in $F \setminus F_y$. If all the fixed points in one of the half-planes delimited by the $y$-axis are saturated or oversaturated, then the robots shall have an agreement on the positive direction of the $x$-axis. So assume that unsaturated fixed points are present in both the half-planes. In this case, the robots select two target fixed points, one from each of the half-planes. Let $f_j$ and $f_u$ be the unsaturated fixed points, which have the highest configuration rank in their respective half-planes. The robots select $f_j$ and $f_u$ as the target fixed points. Note that $f_j$ and $f_u$ may be symmetric images of each other.

### 4.3. Candidate R Selection

This is the algorithm by which the robots select a candidate robot for a target fixed point. Let $f_j$ be the target fixed point. The robots consider the following cases:

1. There exists a robot position which lies within $\rho$ distance from $f_j$. Let $r_i \in R_\rho$ be the closest robot from $C(f_j, \rho)$. The robots select $r_i$ as the candidate robot for $f_j$. If there are multiple such robots, then the robots select the one which has the highest configuration rank.

2. There does not exist a robot position which lies within $\rho$ distance from $f_j$. Let $r_i$ be the closest robot from $f_j$, which does not lie on a saturated circle. The robots select $r_i$ as the candidate robot for $f_j$. If there are multiple such robots, then the robots select the one, which has the highest configuration rank. Note that $r_i$ might lie on an oversaturated circle.

Note that, if $f_j$ lies on the $y$-axis, and $C(t)$ does not have an agreement on the $x$-axis, then there may be two robots (say $r_1$ and $r_2$) having the same configuration rank, which are closest from $f_j$ (case 2) or closest from $C(f_j, \rho)$ (case 1). In case, the configuration is asymmetric, let $r_k$ be a robot position, which does not have a symmetric image about the $y$-axis. If there are multiple such robots, then the robots select the one, which has the highest configuration rank. The candidate robot is selected, from the half-plane, which contains $r_k$. Otherwise, both $r_1$ and $r_2$, are selected as the candidate robots. In case, $f_j$ lies in a half-plane and $C(t)$ does not have an agreement on the $x$-axis, then the candidate robot is selected from the same half-plane in which it belongs.

### 4.4. Moveto Destination

This is the algorithm by which a candidate robot $r_i$ computes its destination point $q(t)$ on the circle centered at its target fixed point $f_j$ and the movement path $P$ along which it will move towards its destination point. The pseudocode of this algorithm is given in Algorithm 1. Let $p(t)$ denote the intersection point between $C(f_j, \rho)$ and $\overline{r_i(t)f_j}$. During its movement towards the circle centered at its target fixed point $f_j$, a candidate robot must avoid collision with the other robots. In order to ensure collision-free movement, a candidate robot considers the following cases:

---

**Algorithm 1:** *MovetoDestination*$(C(t), f_j, r_i)$

---

**Input:** $C(t), f_j, r_i$
**Output:** Movement path $P$ and destination point $q(t)$

1  **if** $d(r_i(t), f_j) < \rho$ **then**
2     Let $l_{ji}(t)$ be the line segment from $f_j$ to $C(f_j, \rho)$, passing through $r_i$;
3     Let $q$ be the intersection point between $l_{ji}(t)$ and $C(f_j, \rho)$;
4     **if** *q is not a robot position* **then**
5         $r_i$ selects $P = \overline{r_i q}$ and $q(t) = q$;
6         $r_i$ starts moving towards $q$ along $\overline{r_i q}$;
7     **else**
8         **if** *there does not exist any robot positions on $C(f_j, \rho)$ other than being collinear with $r_i$ and $f_j$* **then**
9             Let $B_1$ be the ray starting from $r_i(t)$ such that $\angle l_{ji}(t) r_i(t) B_1 = \frac{\pi}{4}$;
10            Let $q_1$ be the intersection point between $C(f_j, \rho)$ and $B_1$;
11            $r_i$ selects $P = \overline{r_i q_1}$ and $q(t) = q_1$;
12            $r_i$ starts moving towards $q_1$ along $\overline{r_i q_1}$;
13         **else**
14            Let $r_u$ be the robot on $C(f_j, \rho)$ such that $\angle \overline{r_i(t)q} r_i(t) \overline{r_i(t) r_u(t)}$ is smallest;
15            Let $B_2$ be the ray starting from $r_i(t)$ such that
               $\angle \overline{r_i(t)q} r_i(t) B_2 = \frac{1}{2} min(\frac{\pi}{2}, \angle \overline{r_i(t)q} r_i(t) \overline{r_i(t) r_u(t)})$;
16            Let $q_2$ be the intersection point between $C(f_j, \rho)$ and $B_2$;
17            $r_i$ selects $P = \overline{r_i q_2}$ and $q(t) = q_2$;
18            $r_i$ starts moving towards $q_2$ along $\overline{r_i q_2}$;
19         **end**
20     **end**
21 **else**
22     Let $p(t)$ be the intersection point between $C(f_j, \rho)$ and $\overline{r_i f_j}$;
23     **if** $\overline{r_i f_j}$ *does not cut any saturated circle* **then**
24         **if** *$p(t)$ is not a robot position* **then**
25            $r_i$ selects $P = \overline{r_i p(t)}$ and $q(t) = p(t)$;
26            $r_i$ starts moving towards $p(t)$ along $\overline{r_i p(t)}$;
27         **else if** *there does not exist any robot positions on $C(f_j, \rho)$ other than being collinear with $r_i$ and $f_j$*
           **then**
28            Let $t^a$ be one of the tangents from $r_i$ to $C(f_j, \rho)$;
29            Let $t^a$ intersects $C(f_j, \rho)$ at $q$;
30            $r_i$ selects $P = \overline{r_i q}$ and $q(t) = q$;
31            $r_i$ starts moving towards $q$ along $\overline{r_i q}$;
32         **else**
33            Let $r_k$ be the robot position on $C(f_j, \rho)$ such that $\angle \overline{r_i(t) r_k(t)} r_i(t) \overline{r_i(t) f_j}$ is the smallest;
34            Let $B_1$ be the ray starting from $r_i(t)$ such that
$$\angle \overline{r_i(t) r_k(t)} r_i(t) B_1 = \frac{1}{2} \angle \overline{r_i(t) r_k(t)} r_i(t) \overline{r_i(t) f_j};$$
35            Let $q_1$ be the intersection point between $C(f_j, \rho)$ and $B_1$;
36            $r_i$ selects $P = \overline{r_i q_1}$ and $q(t) = q_1$;
37            $r_i$ starts moving towards $q_1$ along $\overline{r_i q_1}$;
38         **end**
39     **else**
40         Let $C(f_u, \rho)$ be the first saturated circle which $r_i$ cuts while moving along $\overline{r_i f_j}$;
41         Let $q$ be the intersection point between $\overline{r_i f_j}$ and $C(f_u, \rho)$ which is at closest distance from $r_i$;
42         **if** *q is not a robot position* **then**
43            $r_i$ selects $P = \overline{r_i q}$ and $q(t) = q$;
44            $r_i$ starts moving towards $q$ along $\overline{r_i q}$;
45         **else**
46            Let $r_k$ be the robot on $C(f_u, \rho)$ such that $\angle \overline{r_i(t) f_j} r_i(t) \overline{r_i(t) r_k(t)}$ is the smallest;
47            Let $B_1$ be the ray from $r_i(t)$ such that
$$\angle \overline{r_i(t) f_j} r_i(t) B_1 = \frac{1}{2} min(\angle \overline{r_i(t) f_j} r_i(t) t^a, \angle \overline{r_i(t) f_j} r_i(t) \overline{r_i(t) r_k(t)});$$
48            Let $q_1$ be the intersection point between $B_1$ and $C(f_u, \rho)$ which is at closest distance from
           $r_i$;
49            $r_i$ selects $P = \overline{r_i q_1}$ and $q(t) = q_1$;
50            $r_i$ starts moving towards $q_1$ along $\overline{r_i q_1}$;
51         **end**
52     **end**
53 **end**

1. $d(r_i(t), f_j) > \rho$ and $\overline{r_i(t)f_j}$ does not cut any saturated circle. If $p(t)$ is not a robot position, then $r_i$ selects $q(t) = p(t)$ and $P = \overline{r_i(t)p(t)}$ (Figure 3a). Next, consider the case when $p(t)$ is a robot position and there are no other robot positions on $C(f_j, \rho)$ other than those collinear with $r_i$ and $f_j$. In this case, $r_i$ selects one of the tangent lines to $C(f_j, \rho)$ from its position (say $t^a$) as its movement path. Let $t^a$ intersect $C(f_j, \rho)$ at $q$. In this case $q$ cannot be a robot position. Since $r_i$ is a candidate robot, the line segement $\overline{r_i(t)q}$ cannot possibly contain any robot positions other than $r_i(t)$. It selects $P = t^a$ and $q(t) = q$ (Figure 3b). Otherwise, among the robot positions on $C(f_j, \rho)$ which are not collinear with $r_i$ and $f_j$, let $r_k$ be the robot such that the angle $\angle \overline{r_i(t)f_j}r_i(t)\overline{r_i(t)r_k(t)}$ is smallest. Let $B_1$ be the angle bisector such that $\angle \overline{r_i(t)f_j}r_i(t)B_1 = \frac{1}{2}\angle \overline{r_i(t)f_j}r_i(t)\overline{r_i(t)r_k(t)}$. Note that $B_1$ intersects $C(f_j, \rho)$ at exactly two points. Between these two points, let $q_1$ be the closest point from $r_i$. By the choice of $r_k$, $q_1$ cannot be a robot position. Furthermore, since $r_i$ is a candidate robot, the line segment $\overline{r_i(t)q_1}$ cannot possibly contain any robot positions other than $r_i(t)$. It selects $q(t) = q_1$ and $P = \overline{r_i(t)q_1}$ (Figure 4).

2. $d(r_i(t), f_j) > \rho$ and $\overline{r_i(t)f_j}$ cuts some saturated circle. Let $C(f_u, \rho)$ be the first saturated circle, which $r_i$ cuts while moving along $\overline{r_i(t)f_j}$. Notice that $\overline{r_if_j}$ would intersect $C(f_u, \rho)$ at two points. Consider $q$ to be the intersection point between $C(f_u, \rho)$ and $\overline{r_i(t)f_j}$, which is at the closest distance from $r_i$. Since $r_i$ is a candidate robot, the line segment $\overline{r_i(t)q}$ (excluding point $q$) cannot possibly contain any robot positions other than $r_i(t)$. However, since $q$ is a point on $C(f_u, \rho)$, it may be a robot position. If $q$ is not a robot position, then $r_i$ selects $q(t) = q$ and $P = \overline{r_i(t)q}$ (Figure 5). Otherwise, let $r_k$ (not collinear with $r_i$ and $f_j$) be the robot on $C(f_u, \rho)$ such that angle between $\overline{r_i(t)f_j}$ and $\overline{r_i(t)r_k(t)}$ is the smallest. Since $C(f_u, \rho)$ is saturated, such a robot position always exists on it. Let $B_1$ be the angle bisector, such that $\angle \overline{r_i(t)f_j}r_i(t)B_1 = \frac{1}{2}min(\angle \overline{r_i(t)f_j}r_i(t)t^a, \angle \overline{r_i(t)f_j}r_i(t)\overline{r_i(t)r_k(t)})$. Note that $B_1$ intersects $C(f_u, \rho)$ at exactly two points. Between these two points, let $q_1$ be the closest point from $r_i$. By the choice of $r_k$, $q_1$ cannot be a robot position. Furthermore, since $r_i$ is a candidate robot $\overline{r_i(t)q_1}$ cannot possibly contain any robot positions other than $r_i(t)$. Robot $r_i$ selects $P = \overline{r_i(t)q_1}$ and $q(t) = q_1$ (Figure 6). Note that the choice of $B_1$ ensures that $r_i$ always moves towards $C(f_j, \rho)$.

3. $d(r_i(t), f_j) < \rho$. Let $l_{ji}(t)$ be the line segment from $f_j$ to $C(f_j, \rho)$, passing through $r_i$. Let $q$ be the intersection point between $l_{ji}(t)$ and $C(f_j, \rho)$. Since $r_i$ is a candidate robot, the line segment $\overline{r_i(t)q}$ (excluding point $q$) cannot possibly contain any robot positions other than $r_i(t)$. However, since $q$ is a point on $C(f_j, \rho)$, it may be a robot position. If $q$ is not a robot position, then $r_i$ selects $q(t) = q$ and $P = \overline{r_i(t)q}$ (Figure 7a). Next, consider the case when $q$ is a robot position and $C(f_j, \rho)$ does not contain any robot positions other than being collinear with $r_i$ and $f_j$. Let $B_1$ be the ray starting from $r_i(t)$ such that $\angle \overline{r_i(t)q}r_i(t)B_1 = \frac{\pi}{4}$ (Figure 7b). Suppose $B_1$ intersects $C(f_j, \rho)$ at $q_1$. The candidate robot $r_i$ selects $q(t) = q_1$ and $P = \overline{r_i(t)q_1}$. Otherwise, let $r_u$ (not collinear with $r_i$ and $f_j$) be the robot position on $C(f_j, \rho)$ such that $\angle \overline{r_i(t)q}r_i(t)\overline{r_i(t)r_u(t)}$ is the smallest. Let $B_2$ be the ray starting from $r_i(t)$ such that $\angle \overline{r_i(t)q}r_i(t)B_2 = \frac{1}{2}min(\frac{\pi}{2}, \angle \overline{r_i(t)q}r_i(t)\overline{r_i(t)r_u(t)})$. Suppose $q_2$ is the intersection point between $B_2$ and $C(f_j, \rho)$. The candidate robot selects $q(t) = q_2$ and $P = \overline{r_i(t)q_2}$ (Figure 7c).

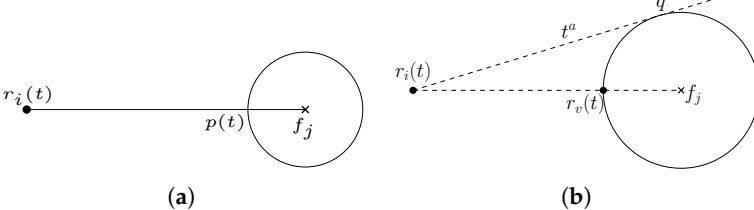

**(a)**                              **(b)**

**Figure 3.** (**a**) $P = \overline{r_i(t)p(t)}$ and $q(t) = p(t)$. (**b**) $r_v(t)$ is the robot position on $p(t)$. $q(t) = q$ and $P = \overline{r_i(t)q}$, where $q$ is the point of intersection between $t^a$ and $C(f_j, \rho)$.

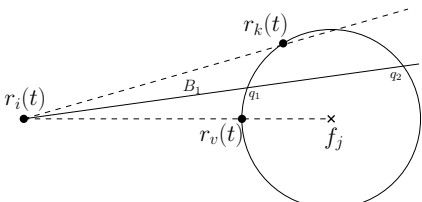

**Figure 4.** $B_1$ is the angle bisector of $\angle \overline{r_i(t)f_j}r_i(t)\overline{r_i(t)r_k(t)}$. It intersects $C(f_j, \rho)$ at $q_1$ and $q_2$. In this case, $r_i$ selects $P = \overline{r_i(t)q_1}$ and $q(t) = q_1$.

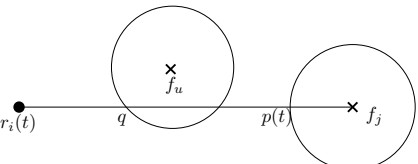

**Figure 5.** $P = \overline{r_i(t)q}$ and $q(t) = q$, where $q$ is the point of intersection between $\overline{r_i(t)f_j}$ and $C(f_u, \rho)$.

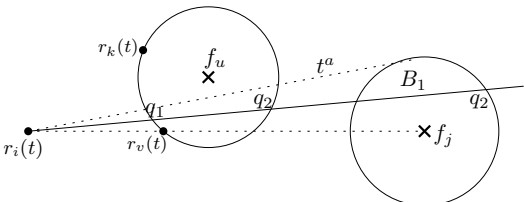

**Figure 6.** $B_1$ is the angle bisector of $\angle \overline{r_i(t)f_j}r_i(t)t^a$. It intersects $C(f_u, \rho)$ at $q_1$ and $q_2$. In this case, $r_i$ selects $P = \overline{r_i(t)q_1}$ and $q(t) = q_1$.

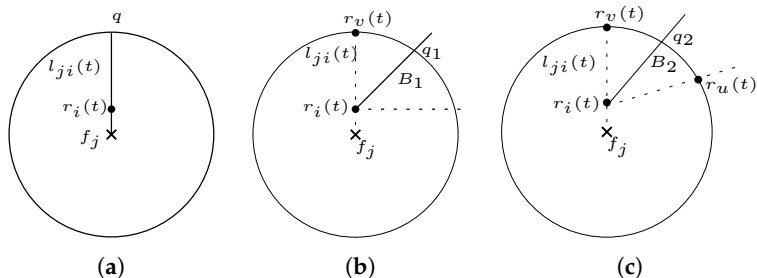

**(a)**                    **(b)**                    **(c)**

**Figure 7.** (**a**) $P = \overline{r_i(t)q}$ and $q(t) = q$, where $q$ is the intersection point between $l_{ji}(t)$ and $C(f_j, \rho)$. (**b**) $q = r_v(t)$. $B_1$ is the ray starting from $r_i(t)$ such that $\angle \overline{r_i(t)r_v(t)}r_i(t)B_1 = \frac{\pi}{4}$. $P = \overline{r_i(t)q_1}$ and $q(t) = q_1$, where $q_1$ is the intersection point between $B_1$ and $C(f_j, \rho)$. (**c**) $q = r_v(t)$. $B_2$ is the ray starting from $r_i(t)$ such that $\angle \overline{r_i(t)r_v(t)}r_i(t)B_2 = \frac{1}{2}\angle \overline{r_i(t)r_v(t)}r_i(t)\overline{r_i(t)r_u(t)}$. $P = \overline{r_i(t)q_2}$ and $q(t) = q_2$, where $q_2$ is the intersection point between $B_2$ and $C(f_j, \rho)$.

In case there are exactly two candidate robots, which lie in different half-planes, each of them computes its destination point and movement path in such a way that, during its movement, it does not cross the $y$-axis. For example, consider the case when the target

fixed point lies on the $y$-axis. A candidate robot will consider the tangent line and robot positions, which lie in its half-plane, while computing its destination point and movement path.

### 4.5. Algorithm One Axis

This is the proposed algorithm that solves the *k*-circle formation problem with one axis agreement. The pseudocode is given in Algorithm 2. Given a configuration $C(t)$, each active robot executes *AlgorithmOneAxis*$(C(t))$. During an execution of *AlgorithmOneAxis*$(C(t))$, if $C(t)$ is not a final configuration, then an active robot (say $r_k$) executes algorithm *AgreementOneAxis*$(C(t))$. Next, $r_k$ considers the following cases:

---

**Algorithm 2:** *AlgorithmOneAxis*

---

    **Input:** $C(t) = (R(t), F)$

1  Let $r_k$ be an active robot at time $t$;
2  $r_k$ executes *AgreementOneAxis*$(C(t))$;
3  **if** *the robots have an agreement on the positive direction of the x-axis* **then**
4      $r_k$ executes *TargetFPSelection*$(C(t))$;
5      Let $f_j$ be the target fixed point;
6      $r_k$ executes *CandidateRSelection*$(C(t), f_j)$;
7      Let $r_i$ be the candidate robot;
8      **if** $r_k = r_i$ **then**
9         |  $r_k$ executes *MovetoDestination*$(C(t), f_j, r_k)$;
10    **end**
11 **else**
12    **if** *all the fixed points in $F \setminus F_y$ are saturated* **then**
13      $r_k$ executes *TargetFPSelection*$(C(t))$;
14      Let $f_j$ be the target fixed point;
15      $r_k$ executes *CandidateRSelection*$(C(t), f_j)$;
16      **if** *there is a unique candidate robot* **then**
17         Let $r_i$ be the candidate robot;
18         **if** $r_k = r_i$ **then**
19            |  $r_k$ executes *MovetoDestination*$(C(t), f_j, r_k)$;
20         **end**
21      **else**
22         Let $r_i$ be the candidate robot such that $r_k$ and $r_i$ lie in the same half-plane;
23         **if** $r_k = r_i$ **then**
24            |  $r_k$ executes *MovetoDestination*$(C(t), f_j, r_k)$;
25         **end**
26      **end**
27    **else**
28      $r_k$ executes *TargetFPSelection*$(C(t))$;
29      Let $f_j$ and $f_b$ be the target fixed points;
30      $r_k$ executes *CandidateRSelection*$(C(t), f_j)$ and *CandidateRSelection*$(C(t), f_b)$;
31      Let $r_i$ and $r_a$ be the candidate robots of $f_j$ and $f_b$, respectively;
32      **if** $r_k = r_i$ **then**
33         |  $r_k$ executes *MovetoDestination*$(C(t), f_j, r_k)$;
34      **else if** $r_k = r_a$ **then**
35         |  $r_k$ executes *MovetoDestination*$(C(t), f_b, r_k)$;
36      **end**
37    **end**
38 **end**

---

1. The robots have an agreement on the positive direction of the $x$-axis. Robot $r_k$ executes *TargetFPSelection*$(C(t))$. In this case there is a unique target fixed point. Let $f_j$ be the target fixed point. Next, $r_k$ identifies the candidate robot by executing *CandidateRSelection*$(C(t), f_j)$. Let $r_i$ be the candidate robot selected for $f_j$. If $r_k = r_i$, then it executes *MovetoDestination*$(C(t), f_j, r_i)$.

2. The robots do not have any agreement on the positive direction of the $x$-axis. Robot $r_k$ considers the following cases:

    (a)    All the fixed points in $F \setminus F_y$ are saturated. Robot $r_k$ executes *TargetFPSelection*$(C(t))$. In this case the unique target fixed point lies on the $y$-axis. Let $f_j$ be the target

fixed point. Robot $r_k$ executes $CandidateRSelection(C(t), f_j)$. Let $r_i$ be the candidate robot. Note that there may be two candidate robots for $f_j$. In that case, suppose $r_i$ is the candidate robot, that lies in the same half-plane containing $r_k$. If $r_k = r_i$, then it executes $MovetoDestination(C(t), f_j, r_i)$.

(b)     There exists an unsaturated fixed point in $F \setminus F_y$. Note that such unsaturated fixed points are present in both the half-planes. Otherwise the robots would have an agreement on the positive direction of the $x$-axis. Robot $r_k$ executes $TargetFPSelection(C(t))$. In this case there are two target fixed points, one from each of the half-planes. Let $f_j$ and $f_u$ be the two target fixed points. Without loss of generality, assume that $r_k$ and $f_j$ lie in the same half-plane. Next, $r_k$ executes $CandidateRSelection(C(t), f_j)$. Let $r_i$ be the candidate robot selected for $f_j$. If $r_k = r_i$, then it executes sub-procedure $MovetoDestination(C(t), f_j, r_i)$.

## 5. Correctness

**Lemma 1.** *Given a configuration $C(t)$ for some $t \geq 0$, if the robots agree upon the positive direction of the $x$-axis, by the execution of $AgreementOneAxis(C(t))$, then the agreement remains invariant at any arbitrary point of time $t' > t$.*

**Proof of Lemma 1.** Let the robots agree upon the positive direction of the $x$-axis, by the execution of $AgreementOneAxis(C(t))$. Consider the following cases:

**Case 1.** $C(t) \in \mathcal{I}_1$, i.e., $F$ is asymmetric about the $y$-axis. Since this agreement is w.r.t. the fixed points, it remains invariant for any $t' > t$.

**Case 2.** $C(t) \in \mathcal{I}_2$ and $C(t)$ is unbalanced. In this case, the agreement on the direction of the positive $x$-axis is based upon robot positions. If the robots move across the $y$-axis from the negative side to the positive side, then the agreement does not change as the positive side of the $y$-axis would still contain the maximum number of robots. During an execution of $TargetFPSelection(C(t))$, the unsaturated fixed points with a higher configuration rank are given preference over the unsaturated fixed points with a lower configuration rank. As a result, the robots move across the $y$-axis from the positive side to the negative side, only when all the fixed points on the positive side of the $y$-axis are either saturated or oversaturated. Due to this movement, the configuration would transform into a balanced configuration. Next, case 3 would follow.

**Case 3.** $C(t) \in \mathcal{I}_2$ is a balanced configuration and all the fixed points in one of the half-planes are either saturated or oversaturated. Notice that a candidate robot, selected by the execution of $CandidateRSelection(C(t))$, would never lie on a saturated circle. As a result, once a circle becomes saturated, it would never become unsaturated. Thus, all the fixed points on the positive side of the $y$-axis would never become unsaturated. This implies that at any $t' > t$ the agreement on the positive direction of the $x$-axis remains invariant.

**Case 4.** $C(t) \in \mathcal{I}_2$ is a balanced configuration with at least one unsaturated fixed point in both the half-planes. Furthermore, $k$ is odd and $F_y \neq \varnothing$. In this case, the positive $x$-axis direction is considered towards the half-plane in which there has been more progress at time $t$. During an execution of $TargetFPSelection(C(t))$, the unsaturated fixed points with higher configuration rank are given preference over the unsaturated fixed points with lower configuration rank. As a result, it is guaranteed to have more progress in the positive side of the $y$-axis for any $t' > t$. Therefore, for any $t' > t$ the agreement on the positive direction of the $x$-axis remains invariant. In case $t = 0$, it might be possible that both the half-planes have the same progress. Since $C(0)$ is asymmetric about the $y$-axis in this case, there exists at least one robot asymmetric robot position. The positive $x$-axis direction is considered towards the half-plane, which contains the asymmetric robot with the highest configuration rank. For any $t' > t$, either $C(t') = C(0)$ or it is guaranteed to have more

progress in the positive side of the $y$-axis. Therefore, the agreement on the positive direction of the $x$-axis remains invariant.

Hence, if the robots agree upon the positive direction of the $x$-axis by the execution of *AgreementOneAxis*($C(t)$), then at any arbitrary point of time $t' > t$ the agreement remains invariant. □

Next, we consider the balanced configurations in which the robots make an agreement on the positive direction of the $x$-axis at some $t' > 0$. Lemma 1 ensures that the agreement remains invariant for any $t'' > t'$. Note that, at any arbitrary point of time $t \in [0, t')$, the robots have selected two target fixed points, one from each of the half-planes. Since the scheduler is assumed to be asynchronous, it is possible to have a candidate robot on the negative side of the $y$-axis, selected at some $t \in [0, t')$ and which has not reached its destination point at $t'$. We need to ensure that there would not be any collision due to the inherent motion of such a candidate robot.

**Lemma 2.** *Let $C(t')$ for some $t' > 0$, be the configuration in which the robots make an agreement on the positive direction of the $x$-axis. Let $r_i$ be the candidate robot on the negative side of the $y$-axis, that was selected for some target fixed point $f_j$ at $t \in [0, t')$. If $t''$ is the point of time at which it re-computes its destination point, then it does not collide with any other candidate robots in the time interval $[t', t'']$.*

**Proof of Lemma 2.** Let $f_a$ be the target fixed point at some $t \in [t', t'']$. Since the robots have agreement on the positive direction of the $x$-axis, a unique candidate robot would be selected by the execution of *CandidateRSelection*($C(t), f_a$). Let $r_b$ be the candidate robot. Note that, $f_a \geq f_j$, i.e., the configuration rank of $f_j$ cannot be higher than $f_a$. Otherwise, $f_j$ would have been selected as the target fixed point. Consider the following cases:

**Case 1.** $f_a = f_j$. In this case $r_a = r_i$. This is because $r_i$ is the candidate robot that was selected for $f_j$ at $t \in [0, t')$ and has not reached $C(f_j, \rho)$. It would remain as the closest robot position from $f_j$, that does not lie on a saturated circle. Since $r_i$ would be the unique robot which is in motion within $d(f_j, r_i)$ distance from $f_j$, there would not be any collision of robots.

**Case 2.** So we assume that $f_j \neq f_a$. The movement paths of $r_i$ and $r_b$ would not intersect. Otherwise, by triangle inequality $r_i$ would have been at closer distance from $f_a$. So $r_i$ would have selected as the candidate robot for $f_a$ by the execution *CandidateRSelection*($C(t), f_a$). Since the movement paths do not intersect, $r_i$ would not collide with $r_b$ during the time interval $[t', t'']$. Since the scheduler is assumed to be asynchronous, it is possible that $r_i$ becomes the candidate robot for $f_a$ as in Figure 8. As the movement paths do not intersect, $r_i$ would continue its movement towards $C(f_j, \rho)$ without collision unless it stops and re-computes its destination point. If it stops it will execute *MovetoDestination*($C(t), f_a, r_i$). It computes its movement path towards $C(f_a, \rho)$ that does not intersect with the movement path of $r_b$. As a result, it would continue its movement towards $C(f_a, \rho)$ in subsequent time without any collision with $r_b$.

Hence, $r_i$ would not collide with any other candidate robots in the time interval $[t', t'']$. □

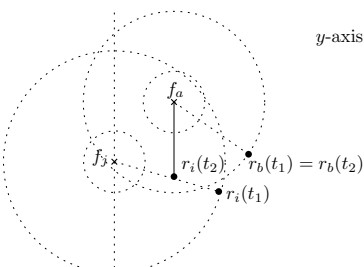

**Figure 8.** Robot $r_i$ has moved from $r_i(t_1)$ to $r_i(t_2)$. It becomes a candidate robot for $f_a$ at time $t_2$.

Theorem 1 characterizes all the configurations and the values of $k$ for which the $k$-circle formation problem is deterministically unsolvable. For some $k > 0$, if the $k$-circle formation problem is deterministically solvable for a given $C(0)$, the robots can identify it in its look phase. The robots must ensure that such configurations would not transform into an configuration that would satisfy the unsolvability criterion (Theorem 1) for any $t > 0$ during an execution of *AlgorithmOneAxis*.

**Lemma 3.** *Given $k > 0$ and $C(0)$, if the $k$-circle formation problem is deterministically solvable, then at any arbitrary point of time $t > 0$ the configuration would not satisfy the unsolvability criterion (Theorem 1).*

**Proof of Lemma 3.** Since the $k$-circle formation problem is deterministically solvable for every even value of $k$, we assume that $k$ is odd. Note that all the initial configurations, in which $F$ is asymmetric about the $y$-axis or in which $F_y = \varnothing$, would never satisfy the unsolvability criterion stated in Theorem 1. So we only need to consider all the initial configurations in which $F$ is symmetric about the $y$-axis and $F_y \neq \varnothing$. So, $C(0) \notin \mathcal{I}_1 \cup \mathcal{I}_4$. Furthermore, $C(0) \notin \mathcal{I}_5$ (Otherwise, initially it would have been unsolvable). Therefore, $C(0) \in \mathcal{I}_2 \cup \mathcal{I}_3$. We have the following cases:

**Case 1.** The robots make an agreement on the positive direction of $x$-axis, which remains invariant for any $t > 0$ (Lemma 1). Since the agreement remains invariant, even if the configuration becomes symmetric about the $y$-axis, the configuration will not satisfy the unsolvability criterion stated in Theorem 1 for any $t > 0$.

**Case 2.** The robots decide to transform $C(0)$ into an unbalanced configuration, in order to make an agreement on the positive direction of $x$-axis. This includes the following configurations:

1. $C(0) \in \mathcal{I}_3$.
2. $C(0) \in \mathcal{I}_2$ and it is balanced with at least one unsaturated fixed point in both the half-planes and $R_y(t) \neq \varnothing$.

Let $t'$ be earliest possible point of time at which it becomes unbalanced. In the time interval 0 to $t'$, only the topmost robot on the $y$-axis would move along the $y$-axis. As a result, the configuration would not satisfy the unsolvability criterion (Theorem 1) for any $t \in [0, t')$. At $t'$, the robots make an agreement on the positive direction of $x$-axis. Next, the proof follows from case 1.

Therefore, $C(0)$ would not transform into an unsolvable configuration at any arbitrary point of time $t > 0$. □

Given a configuration $C(t)$, let $n_k(t)$ denote the number of unsaturated fixed points. The robots may select one or two target fixed points. First, consider the case when the target fixed point is unique. Suppose, $f_j$ is the target fixed point and $r_i$ its candidate robot selected by the robots. Let $P$ and $q(t)$ be the movement path and destination point, respectively, computed by $r_i$ at time $t$, by the execution of *MovetoDestination*$(C(t), f_j, r_i)$. Consider a straight line along $P$ towards $C(f_j, \rho)$ intersecting the circle $C(f_j, \rho)$ first at $s(t)$ (The line

would always intersect $C(f_j, \rho))$ at time $t$. Suppose $d_j(t)$ denotes the distance between $r_i(t)$ and $s(t)$. Let $N_i(t)$ denote the deficit in the number of robots in order to make $f_i$ a saturated fixed point. Let $V_j(t) = (n_k(t), N_j(t), d_j(t))$.

We say that there has been significant progress in the time interval $t$ to $t'$ if $V_j(t') < V_j(t)$, i.e., one of the following conditions holds:

1. $n_k(t') < n_k(t)$, or
2. $n_k(t') = n_k(t)$ and $N_j(t') < N_j(t)$, or
3. $n_k(t') = n_k(t)$ and $N_j(t') = N_j(t)$ and $d_j(t') + \delta \le d_j(t)$.

**Lemma 4.** *Let $t'$ be an arbitrary point of time before $r_i$ reaches its destination computed at time $t$. During an execution of $AlgorithmOneAxis(C(t))$, execution of $MovetoDestination(C(t), f_j, r_i)$ ensures that $d_j(t') + \delta \le d_j(t)$.*

**Proof of Lemma 4.** Let $P$ and $P'$ be the selected movement paths for $r_i$ at time $t$ and $t'$, respectively. We have $d_j(t) = d(r_i(t), s(t))$ and $d_j(t') = d(r_i(t'), s(t'))$. Note that $q(t) = s(t)$ implies that the destination point lies on $C(f_j, \rho)$. Consider the following cases:

**Case 1.** $q(t) = s(t)$ and $p(t)$ does not contain any robot position. This is the case where the robot moves straight towards $f_j$, i.e., $P = \overline{r_i(t)f_j}$ and the destination point $q(t)$ lies on $C(f_j, \rho)$ (Step 25 of Algorithm 1). At time $t'$ there would not be any robot on $q(t)$ and $r_i$ would continue along the same path. Since $\delta$ is the minimum displacement in a round, $d_j(t') + \delta \le d_j(t)$. Recall that $p(t)$ denotes the intersection point between $C(f_j, \rho)$ and $\overline{r_if_j}$. The movements are shown in Figure 9.

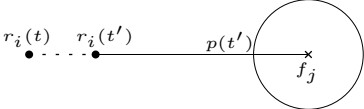

**Figure 9.** Robot $r_i$ has moved from $r_i(t)$ to $r_i(t')$, along $P = \overline{r_i(t)p(t)}$ towards $q(t) = p(t)$ computed at time $t$. Robot $r_i$ selects $P' = \overline{r_i(t')p(t')}$ and $q(t') = p(t')$ at time $t'$. In this case, $q(t') = q(t)$. Furthermore, $q(t) = s(t)$ and $q(t') = s(t')$, i.e., the destination point lies on $C(f_j, \rho)$.

**Case 2.** $q(t) = s(t)$ and $p(t)$ contains a robot position. There are robot positions on $C(f_j, \rho)$, that are not collinear with $r_i$ and $p(t)$. By step 36 of Algorithm 1 robot $r_i$ computes the movement path $P$ and destination point $q(t)$. It starts moving towards $q(t)$ along $P$. At time $t' > t$, let $s(t')$ be the intersection point between $C(f_j, \rho)$ and $\overline{r_i(t')f_j}$. Note that, $p(t')$ is not a robot position. Robot $r_i$ selects $P' = \overline{r_i(t')f_j}$ and $q(t') = p(t')$. We have $d(r_i(t'), q(t)) > d(r_i(t'), q(t'))$ and $d(r_i(t), q(t)) - d(r_i(t'), q(t')) > d(r_i(t), q(t)) - d(r_i(t'), q(t)) \ge \delta$. This implies that $d_j(t') + \delta \le d_j(t)$. The movements are shown in Figure 10.

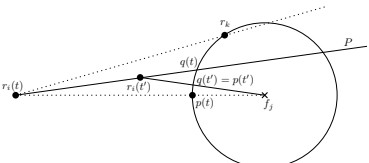

**Figure 10.** Robot $r_i$ has moved from $r_i(t)$ to $r_i(t')$, along $P$ towards $q(t)$ computed at time $t$. Robot $r_i$ selects $P' = \overline{r_i(t')p(t')}$ and $q(t') = p(t')$ at time $t'$. Furthermore, $q(t) = s(t)$ and $q(t') = s(t')$, i.e., the destination point lies on $C(f_j, \rho)$.

**Case 3.** $q(t) = s(t)$ and $p(t)$ contains a robot position. There are no robots on $C(f_j, \rho)$, other than being collinear with $r_i$ and $f_j$. By step 30 of Algorithm 1 robot $r_i$ computes the movement path $P$ and destination point $q(t)$. This case is similar to case 2. The movements are shown in Figure 11.

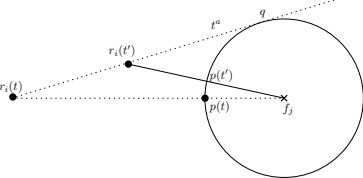

**Figure 11.** Robot $r_i$ has moved from $r_i(t)$ to $r_i(t')$, along $P = \overline{r_i(t)q}$ towards $q(t) = q$ computed at time $t$ ($q$ is the point of intersection between $C(f_j, \rho)$ and $t^a$). Robot $r_i$ selects $P' = \overline{r_i(t')p(t')}$ and $q(t') = p(t')$ at time $t'$. Furthermore, $q(t) = s(t)$ and $q(t') = s(t')$, i.e., the destination point lies on $C(f_j, \rho)$.

**Case 4.** $q(t) \neq s(t)$. In this case $q(t)$ lies on a saturated circle $C(f_u, \rho)$ for some $f_u \neq f_j$. Note that, $C(f_u, \rho)$ is the first circle, that $r_i$ cuts while moving along $\overline{r_i(t)f_j}$. First, consider the case in which $P = \overline{r_i(t)q}$ and $q(t) = q$ (Step 43 of Algorithm 1), where $q$ is intersection point between $\overline{r_i(t)f_j}$ and $C(f_u, \rho)$, which is at closest distance from $r_i$. Since $\delta$ is the minimum displacement in a round, $d_j(t') + \delta \leq d_j(t)$. The movements are shown in (Figure 12). Next, consider the case in which $r_i$ computes its movement path $P$ by step 49 of Algorithm 1. It starts moving towards $q(t)$ along path $P$. At time $t' > t$, let $p'$ be the intersection point between $C(f_u, \rho)$ and $\overline{r_i(t')f_j}$. Note that $p'$ is not a robot position. Robot $r_i$ selects $P' = \overline{r_i(t')f_j}$ and $q(t') = p'$ (Figure 13). We have $d(r_i(t'), s(t)) > d(r_i(t'), s(t'))$ and $d(r_i(t), s(t)) - d(r_i(t'), s(t')) > d(r_i(t), s(t)) - d(r_i(t'), s(t')) \geq \delta$. This implies that $d_j(t') + \delta \leq d_j(t)$.

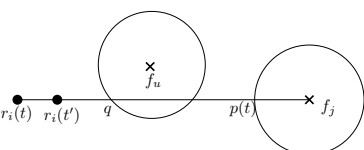

**Figure 12.** $C(f_u, \rho)$ is a saturated circle and $q$ is the point of intersection between $C(f_u, \rho)$ and $\overline{r_i(t)f_j}$, which is at closest distance from $r_i$. Robot $r_i$ has moved from $r_i(t)$ to $r_i(t')$, along $P = \overline{r_i(t)q}$ towards $q(t) = q$ computed at time $t$. Robot $r_i$ selects $P' = \overline{r_i(t')q}$ and $q(t')$ on $C(f_u, \rho)$ at time $t'$. In this case, $q(t') = q(t)$. Furthermore, $q(t) \neq s(t)$ and $q(t') \neq s(t')$, i.e., the destination point does not lie on $C(f_j, \rho)$.

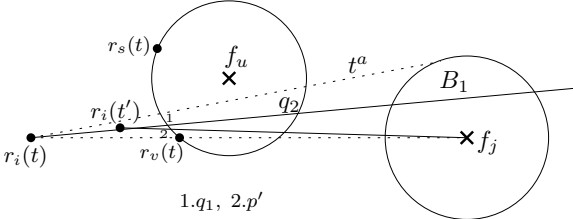

**Figure 13.** Robot $r_i$ has moved from $r_i(t)$ to $r_i(t')$, along $P = \overline{r_i(t)q_1}$ towards $q(t) = q_1$ computed at time $t$. Robot $r_i$ selects $P' = \overline{r_i(t)q}$ and $q(t') = p'$ ($p'$ is the point of intersection between $C(f_u, \rho)$ and $\overline{r_i(t')f_j}$) on $C(f_u, \rho)$ at time $t'$

**Case 5.** $d(r_i, f_j) < \rho$. We have $q(t) = s(t)$. Let $q$ be the intersection point between $C(f_j, \rho)$ and $l_{ji}(t)$. First, consider the case when $r_i$ selects $P = \overline{r_i(t)q}$ and $q(t) = q$ (Step 5 of Algorithm 1). At time $t'$, there would not be any robot position on $q(t)$. Robot $r_i$ selects $P' = \overline{r_i(t')q}$. Since $\delta$ is the minimum displacement in a round, $d_j(t') + \delta \leq d_j(t)$. Movements are shown in Figure 14a. Next, consider the case in which $r_i$ selects its movement path $P$ by step 11 or step 17 of Algorithm 1. We have $d(r_i(t'), q(t)) > d(r_i(t'), q(t'))$ and $d(r_i(t), q(t)) - d(r_i(t'), q(t')) > d(r_i(t), q(t)) - d(r_i(t'), q(t')) \geq \delta$. Hence, $d_j(t') + \delta \leq d_j(t)$. Movements are shown in Figure 14b,c.

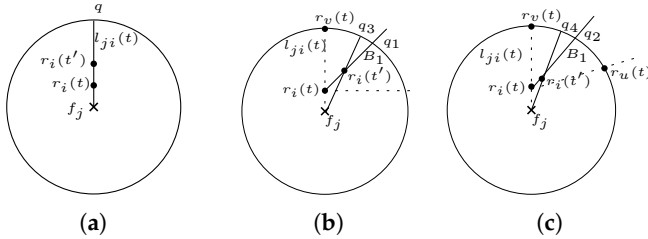

(a)         (b)         (c)

**Figure 14.** (a) Robot $r_i$ has moved from $r_i(t)$ to $r_i(t')$ along $P = \overline{r_i(t)q}$ towards $q(t) = q$ ($q$ is the point of intersection between $C(f_j, \rho)$ and $l_{ji}(t)$). It selects $P' = \overline{r_i(t')q}$ and $q(t') = q$. (b) At time $t$, $r_i$ selects $P = \overline{r_i(t)q_1}$ and $q(t) = q_1$. It selects $P' = \overline{r_i(t)q_3}$ and $q(t') = q_3$. (c) At time $t$, $r_i$ selects $P = \overline{r_i(t)q_2}$ and $q(t) = q_2$. It selects $\overline{r_i(t)q_4}$ and $q(t') = q_4$.

Hence, execution of $MovetoDestination(C(t))$ ensures $d_j(t') + \delta \leq d_j(t)$.    □

**Lemma 5.** *Let $f_j$ be the target fixed point and $r_i$ its candidate robot in the configuration $C(t)$. During an execution of $AlgorithmOneAxis(C(t))$, execution of $MovetoDestination(C(t), f_j, r_i)$ ensures significant progress.*

**Proof of Lemma 5.** Let $r_i$ compute movement path $P$ and destination point $q(t)$ by the execution of $MovetoDestination(C(t), f_j, r_i)$ at time $t$. Let $t' > t$ be an arbitrary point of time at which $r_i$ has completed at least one LCM cycle. We need to show that there has been significant progress in between the time interval $t$ to $t'$. We have the following cases:

**Case 1.** $r_i(t') = q(t)$ and $r_i$ is on the $C(f_j, \rho)$. We have the following two sub-cases:

**Subcase 1.** If $C(f_j, \rho)$ has exactly $k$ robots on it, then $n_k(t') = n_k(t) - 1$, ensuring significant progress.

**Subcase 2.** If $C(f_j, \rho)$ has less than $k$ robots on it, then $N_j(t') = N_j(t) - 1$, ensuring significant progress.

**Case 2.** $r_i(t') \neq q(t)$ and $r_i$ is not on any oversaturated $C(f_u, \rho)$. In this case $d_j(t') + \delta \leq d_j(t)$ by Lemma 4, which ensures significant progress.

**Case 3.** $r_i(t') \neq q(t)$ and $r_i$ is on an oversaturated $C(f_u, \rho)$. Since at this stage, a candidate robot for $f_j$ will be selected again, algorithm $CandidateRSelection(C(t'), f_j)$ will select a robot $r_k$ such that $d(r_k(t'), f_j) \leq d(r_i(t'), f_j)$. Either $r_k = r_i$ or $r_k \neq r_i$. By Lemma 4, significant progress is ensured, in both the cases.

Hence, execution of $MovetoDestination(C(t), f_j, r_i)$ ensures significant progress.    □

**Lemma 6.** *Let $f_j$ be a target fixed point and $r_i$ its unique selected candidate robot at time $t$. Until $r_i$ reaches its destination point computed at time $t$, it remains the candidate robot for $f_j$.*

**Proof of Lemma 6.** Let $r_i$ compute its movement path $P$ and destination point $q(t)$ by the execution of $MovetoDestination(C(t), f_j, r_i)$. Note that, $q(t)$ is either a point on the circle $C(f_j, \rho)$ or on some saturated circle $C(f_u, \rho)$. Let $t'$ be an arbitrary point of time such that $r_i(t') \neq q(t)$. At time $t'$, $f_j$ remains an unsaturated fixed point. As a result, $f_j$ remains a target fixed point at time $t'$. Lemma 4 guarantees that $r_i$ has moved at least $\delta$ amount closer to $C(f_j, \rho)$. Therefore, it remains the candidate robot for $f_j$.    □

Next, we consider the case when there are two candidate robots for a target fixed point. Since robots have an agreement on the directions and orientations of the $y$-axis,

there can be at most two candidate robots at any point of time. Note that, in this case, the configuration would have a unique target fixed point, that lies on the $y$-axis.

**Lemma 7.** *Let $f_j$ be the target fixed point and $r_i$ and $r_v$ are the two selected candidate robots for $f_j$ at time t. Until at least one of them reaches its destination point computed at time t, no other robot becomes a candidate robot. If one of the candidate robots have reached its destination point and the other one has not, then the other robot either continues its inherent motion towards its destination point (computed at time t) without any collision or gets selected as a candidate robot only when $N_j(t)$ reduces by one.*

**Proof of Lemma 7.** Let $t' > t$ be an arbitrary point of time when at least one of the candidate robots has completed its LCM cycle. Without loss of generality, assume that $r_i$ has completed its LCM cycle at $t'$. Let $q(t)$ be the destination point and $P$ be the movement path computed for $r_i$ by *MovetoDestination*$(C(t), f_j, r_i)$. Note that $q(t)$ is a point either on the $C(f_j, \rho)$ or on some saturated $C(f_u, \rho)$. We have the following cases:

**Case 1.** $q(t)$ is a point on the circle $C(f_j, \rho)$. We have the following subcases:

**Subcase 1.** $r_i(t') = q(t)$. Since $r_i$ has reached its destination, the first part of the lemma follows. We have $N_j(t') = N_j(t) - 1$. At $t'$, if $r_v$ has also completed its LCM cycle and has not reached its destination point, then it becomes the next candidate robot for $f_j$. If $r_v$ is in motion, then being the only robot in motion within the annulus region between $C(f_j, \rho)$ and $C(f_j, d(f_j, r_v(t')))$, it continues its motion without any collision. Note that, in this case, no other robot will be selected for movement until $r_v$ reaches its destination.

**Subcase 2.** $r_i(t') \neq q(t)$. First consider that $|d(f_j, r_i(t')) - \rho| > |d(f_j, r_v(t')) - \rho|$, i.e., robot $r_v$ is closer to $C(f_j, \rho)$ than $r_i$. At $t'$, either $r_v$ has also completed its LCM cycle and has not reached its destination point or $r_v$ is in motion. In both the cases, $r_v$ remains a candidate robot for $f_j$. The first part of the lemma follows for $r_v$. Robot $r_i$ will be selected as a candidate robot when $r_v$ will reach $C(f_j, \rho)$. Next consider that $|d(f_j, r_i(t')) - \rho| < |d(f_j, r_v(t')) - \rho|$, i.e., robot $r_i$ is closer to $C(f_j, \rho)$ than $r_v$. Robot $r_i$ will be selected as a candidate robot. At $t'$, if $r_v$ has also completed its LCM cycle, then it will become the candidate robot when $r_i$ will reach $C(f_j, \rho)$. If $r_v$ is in motion, then it continues its motion without any collision (As destination point and movement path computed by $r_i$ and $r_v$, respectively are separated by the $y$-axis and there are no other robots in the half-plane containing $r_v$, which is in motion within the annulus region between $C(f_j, \rho)$ and $C(f_j, d(f_j, r_v(t')))$). We have two possible cases. First, $r_v$ will also reach $C(f_j, \rho)$. Second, if it stops before reaching $C(f_j, \rho)$, then it will become a candidate robot only when $r_i$ will reach $C(f_j, \rho)$.

**Case 2.** $q(t)$ is a point on some saturated circle $C(f_u, \rho)$. We have the following cases:

**Subcase 1.** $r_i(t') = q(t)$. Since $r_i$ has reached its destination, the first part of the lemma follows. At $t'$, since $C(f_u, \rho)$ contains $k + 1$ robots, the next candidate robot for $f_j$ will be selected from $C(f_u, \rho)$. Note that, this robot position would have higher $y$-coordinate than $q(t)$. If $r_v$ has also completed its LCM cycle and has not reached its destination point, then it will become a candidate robot for $f_j$ only when $N_j(t'') = N_j(t') - 1$ for some $t'' > t'$. If $r_v$ is in motion, then it continues its motion without any collision (It is the only robot, which is in motion within the annulus region between $C(f_j, \rho)$ and $C(f_j, d(f_j, r_v(t')))$ and below the point $q(t)$).

**Subcase 2.** $r_i(t') \neq q(t)$. First consider that $|d(f_j, r_i(t')) - \rho| > |d(f_j, r_v(t')) - \rho|$, i.e., robot $r_v$ is closer to $C(f_j, \rho)$ than $r_i$. At $t'$, either $r_v$ has also completed its LCM cycle and has not reached its destination point or $r_v$ is in motion. In both cases, $r_v$ remains a candidate robot for $f_j$. The first part of the lemma follows for $r_v$. Robot $r_i$ will be selected as a candidate robot only when $N_j(t)$ reduces by one. Next consider that $|d(f_j, r_i(t')) - \rho| < |d(f_j, r_v(t')) - \rho|$,

i.e., robot $r_i$ is closer to $C(f_j, \rho)$ than $r_v$. Robot $r_i$ will be selected as a candidate robot. At $t'$, if $r_v$ has also completed its LCM cycle and has not reached its destination point, then it will become a candidate robot only when $N_j(t)$ reduces by one. If $r_v$ is in motion, then it continues its motion without any collision (As destination point and path computed by $r_i$ and $r_v$, respectively, are separated by the $y$-axis and there are no other robots in motion within the annulus region between $C(f_j, \rho)$ and $C(f_j, d(f_j, r_v(t')))$ and below the point $q(t)$). We have two possible cases. First, $r_v$ will also reach $C(f_u, \rho)$. Second, if it stops before reaching $C(f_u, \rho)$, then it will become a candidate robot only when $N_j(t)$ reduces by one. □

Next, we consider the case when there are two target fixed points, one from each half-plane. Let $f_j$ and $f_a$ be the target fixed points at time $t$. Let $r_i$ and $r_b$ be their respective candidate robots. We have $V_j(t) = (n_k(t), N_j(t), d_j(t))$ and $V_a(t) = (n_k(t), N_a(t), d_a(t))$.

**Lemma 8.** *Let $C(t)$ admit two target fixed points during an execution of $AlgorithmOneAxis(C(t))$ and $t' > t$ be an arbitrary point of time when at least one candidate robot has completed its LCM cycle. For at least one target fixed point $f_i \in \{f_j, f_a\}$ and its candidate robot, $d_i(t') + \delta \leq d_i(t)$.*

**Proof of Lemma 8.** Each target fixed point is unique in their respective half-planes. Execution of $AlgorithmOneAxis(C(t))$ ensures that for each target fixed point, its candidate robot is selected from its respective half-planes. The circle formation process continues independently in both the half-planes. This implies that for each $i \in \{j, a\}$, $V_i(t)$ is updated only due to the movement of $f_i$'s candidate robot. Without loss of generality, suppose candidate robot $r_i$ of the target fixed point $f_j$ has completed its LCM cycle. By Lemma 4, $d_j(t') + \delta \leq d_j(t)$ is ensured. □

**Lemma 9.** *Let $C(t)$ admit two target fixed points during an execution of $AlgorithmOneAxis(C(t))$ and $t' > t$ be an arbitrary point of time when at least one candidate robot has completed its LCM cycle. Execution of $AlgorithmOneAxis(C(t))$ ensures significant progress.*

**Proof of Lemma 9.** Lemma 8 ensures that for at least one target fixed point $f_i \in \{f_j, f_a\}$ and its candidate robot, $d_i(t') + \delta \leq d_i(t)$ holds. Without loss of generality, assume that for the target fixed point $f_j$ we have $d_j(t') + \delta \leq d_j(t)$ in the time interval $t$ to $t'$. By Lemma 5, we have $V_j(t') < V_j(t)$, i.e., significant progress is ensured. □

**Theorem 2.** *If the initial configuration $C(0) \in \{\mathcal{I}_1 \cup \mathcal{I}_2 \cup \mathcal{I}_3 \cup \mathcal{I}_4 \cup \mathcal{I}_5\}$ and $C(0)$ does not satisfy the unsolvability criterion stated in Theorem 1, then the robots would eventually solve the k-circle formation problem under one axis agreement, by the execution of $AlgorithmOneAxis$.*

**Proof of Theorem 2.** Lemma 3 guarantees that for any $t > 0$, the configuration $C(t)$ would not satisfy the unsolvability criterion stated in Theorem 1. We have the following cases:

**Case 1.** There is a unique target fixed point (say $f_j$) in the configuration. Lemma 5 ensures that each time a candidate robot gets activated, significant progress is ensured. If there is a unique candidate robot for $f_j$, then Lemma 6 guarantees that until the candidate robot reaches its destination, it would remain the candidate robot. In case there are two candidate robots for $f_j$, then Lemma 7 guarantees that until one of the candidate robots reaches its destination point, no other robot will become a candidate robot. As a result, one of the candidate robots will reach its destination point eventually. If the other candidate robot does not reach its destination point, then it becomes a candidate robot for $f_j$ when $N_j(t)$ reduces by one. Thus, the circle formation process around all the fixed points will be completed eventually.

**Case 2.** There are two target fixed points. Note that the target fixed points lie in different half-planes delimited by the $y$-axis. Lemma 9 ensures significant progress. Lemma 6

guarantees that until a candidate robot reaches its destination, it remains the candidate robot. Note that in this case for each of the target fixed points, always a unique candidate robot gets selected. Thus, the circle formation process around all the fixed points will be completed eventually.

Hence, the robots would eventually solve the *k*-circle formation problem with one axis agreement. □

From Theorem 2, it follows that the robots would solve the *k*-circle formation problem under one axis agreement within finite time. Since we have considered the scheduler to be asynchronous, the robots do not have any common notion of time. As a result, the actual time to solve the *k*-circle formation problem depends upon the scheduling of the robots. We use the notion of an epoch [30] to discuss the runtime complexity of our proposed algorithm. An epoch is the time interval in which all the robots in the configuration have performed their LCM cycles at least once. According to this definition, the time is divided into global epochs. We also assume that the robots have rigid motion, i.e., the robot is guaranteed to reach its destination whenever it moves. In such a setting, we have the following observations:

1. If a candidate robot does not have to pass through a saturated circle in order to reach the circle centered at its target fixed point, then it would reach the circle within one epoch.
2. If a candidate robot has to pass through a saturated circle in order to reach the circle centered at its target fixed point, then it would reach the circle in at most three epochs. This is because the movement path would intersect the saturated circle either one or two times.

From the above two observations, it follows that a candidate robot would reach the circle centered at its target fixed point within $2(m-1)+1 = 2m-1$ epochs. This is because, it might have to pass through $(m-1)$ number of saturated circles. Since *AlgorithmOneAxis* is sequential, each target fixed point would need at most $k(2m-1)$ epochs to become saturated. Therefore, the *k*-circle formation problem would be solved within $\mathcal{O}(m^2k)$ epochs. This is a loose upper bound on the running time of *AlgorithmOneAxis* in terms of epochs.

## 6. Relationship between the k-Circle Formation Problem and the k-epf Problem

Given $m > 0$ fixed points and $n = km$ robots for some positive integer *k*, the *k*-epf problem asks exactly *k* robots to reach and remain in each fixed point. Since the definition of the *k*-circle formation problem asks for distinct robot positions, we only consider the initial configurations with distinct robot positions. We want to prove the following theorem.

**Theorem 3.** *For a given initial configuration with distinct robot positions and a positive integer k, if the k-circle formation problem is deterministically solvable then the k-epf problem is also deterministically solvable.*

In order to prove the above theorem, we modify the proposed algorithm *AlgorithmOneAxis*, to solve the *k*-epf problem deterministically within finite time.

*Algorithm for the k-epf Problem*

Let $C(0)$ be the given initial configuration. Suppose the *k*-circle formation problem has been solved in $C(t)$, for some $t \geq 0$, with radius $\rho$, by the execution of *AlgorithmOneAxis*. In order to solve the *k*-epf problem, the robots must reach the fixed points. The robots can accomplish this by moving in a straight line towards the fixed point. Since the robots are oblivious, they do not remember any information about the past events. Therefore, if any robot stops before reaching the fixed point for some $t' > t$, it would not remember that the *k*-circle formation problem has already been solved. As a result, it will again start executing

*AlgorithmOneAxis*. In order to resolve such a situation, consider the following definition. A configuration is said to satisfy Property 1, if the following conditions hold:

1. Each robot lies within $\rho$ distance from some fixed point.
2. For each $f_i \in F$, there are at most $k$ robots, which lie within $\rho$ distance from $f_i$.

Given a configuration which satisfies Property 1, let $\mathcal{A}$ be an algorithm as follows:

1. If there exists a robot $r_i$ such that $0 < d(r_i, f_j) \leq \rho$ for some $f_j \in F$, then $r_i$ moves along $\overline{r_i f_j}$ towards $f_j$.

Define algorithm *Algokepf* as follows:

1. If the current configuration satisfies Property 1, then execute $\mathcal{A}$.
2. Else the robots execute *AlgorithmOneAxis*.

During an execution of $\mathcal{A}$, it must be ensured that none of the robots have any inherent motion, which is not directed towards the fixed point. Since all the robots are stationary in the initial configuration, if $C(0)$ satisfies Property 1, then none of the robots would have any inherent motion.

**Lemma 10.** *During an execution of AlgorithmOneAxis if $t > 0$ is the earliest possible point of time at which the configuration $C(t)$ satisfies Property 1, then none of the robots would have any inherent motion in $C(t)$.*

**Proof of Lemma 10.** Since $C(t)$ satisfies Property 1, each robot lies within $\rho$ distance from some fixed point. Furthermore, notice that there are no oversaturated circles in $C(t)$. Let $f_j$ be the target fixed point which became saturated at time $t$ due to the movement of a candidate robot (say $r_i$). Notice that if $f_j$ lies on the *y*-axis and the configuration is symmetric, there would be two such candidate robots. In that case, we assume that both of them reached $C(f_j, \rho)$ at time $t$. Otherwise, the configuration $C(t)$ cannot possibly satisfy Property 1. Suppose $r_i$ became a candidate robot at some time $t_1 < t$ by the execution of *CandidateRSelection*. Note that in the time interval $[t_1, t)$, the distance of $r_i$ from $f_j$ was greater than $\rho$. Otherwise, the choice of $t$ is wrong. If there were two candidate robots for $f_j$, then this is true for both the candidate robots. Furthermore, at time $t_1$ there were no robot position (say $r_a$) such that $d(f_j, r_a(t)) < \rho$. Otherwise, $r_a$ would have been selected as a candidate robot. Notice that the candidate robot(s) was the only robot which was moving towards $C(f_j, \rho)$. Therefore, all the robots on $C(f_j, \rho)$ are static at $t$. Next, consider a fixed point $f_l \in F$ such that $f_l$ has higher configuration rank than $f_j$. All the robots within $\rho$ distance from $f_l$ must lie on $C(f_l, \rho)$. This is because, during an execution of *CandidateRSelection* for a fixed point, a robot within $\rho$ distance from that fixed point is given higher preference than any robot at greater than $\rho$ distance from that fixed point. Since $C(f_l, \rho)$ is not oversaturated, all the robots are static at time $t$. Next, consider a fixed point $f_b \in F$ such that $f_b$ has lower configuration rank than $f_j$. By the choice of $f_j$ and $r_i$, none of the robots within $\rho$ distance from $f_b$ were selected as a candidate robot. Therefore, all the robots within $\rho$ distance from $f_b$ are static at time $t$. Hence, if the configuration $C(t)$ satisfies Property 1, then none of the robots have any inherent motion in $C(t)$. $\square$

**Theorem 4.** *If the initial configuration $C(0) \in \{\mathcal{I}_1 \cup \mathcal{I}_2 \cup \mathcal{I}_3 \cup \mathcal{I}_4 \cup \mathcal{I}_5\}$ and $C(0)$ does not satisfy the unsolvability criterion stated in Theorem 1, then the robots would eventually solve the k-epf problem under one axis agreement, by the execution of algorithm Algokepf.*

**Proof of Theorem 4.** First, consider the case when the configuration does not satisfy Property 1. The robots would start executing *AlgorithmOneAxis*. From Theorem 2, it follows that the configuration would eventually satisfy Property 1. Next, consider the case when the configuration satisfies Property 1. From Lemma 10, it follows that all the robots would be static in such a configuration. The robots would start executing $\mathcal{A}$. During an execution of $\mathcal{A}$, each robot moves in a straight line by at least $\delta$ distance, towards the fixed point

from which it is at the closest distance. Since $\rho$ is finite and there are finitely many robots, eventually each of the fixed points will contain exactly $k$ robots.

Hence, the robots would eventually solve the $k$-epf problem by the execution of algorithm *Algokepf*. □

The above theorem provides an evidence that a deterministic distributed algorithm to solve the $k$-circle formation problem can be modified to solve the $k$-epf problem and proves Theorem 3. Notice that during an execution of algorithm *Algokepf*, the robots are only allowed to create a multiplicity on the fixed points. Therefore, the existence of a deterministic distributed algorithm which solves the $k$-epf problem, without allowing a robot multiplicity point outside the fixed points, is guaranteed by Theorem 3.

## 7. Conclusions

This paper studies the $k$-circle formation problem by asynchronous, autonomous, anonymous and oblivious robots in the Euclidean plane. The problem is investigated in a setting where the robots have an agreement on the direction and orientation of the $y$-axis. The following three main results have been proved:

1.  If the initial configuration $C(0)$ is symmetric about the $y$-axis such that $F_y \neq \varnothing$ (there are fixed points on the $y$-axis) and $R_y(0) = \varnothing$ (there are no robot positions on the the $y$-axis), then the $k$-circle formation problem is deterministically unsolvable for odd values of $k$. This is the complete set of the initial configurations and values of $k$ for which the $k$-circle formation problem is deterministically unsolvable under this setting.
2.  For the rest of the configurations and the values of $k$, a deterministic distributed algorithm has been proposed under one axis agreement.
3.  It has also been shown that if the $k$-circle formation problem is deterministically solvable then the $k$-epf problem is also deterministically solvable. This has been established by modifying *AlgorithmOneAxis*; the modified algorithm *Algokepf* deterministically solves the $k$-epf problem.

Future work: The assumption of agreement on the $y$-axis has a strong influence on the results. The natural follow up work would be to consider the problem without this assumption. It is expected that the set of unsolvable initial configurations would increase significantly. Another direction of future work would be to consider the problem with different radii for the circles. The problem can also be considered with two different objectives: (i) minimize the total distance traveled by all the robots; (ii) minimize the maximum distance traveled by a single robot. It is to be noted that there is no guarantee that the problems are solvable with these objectives. Another significant assumption used in this paper is unlimited visibility. This problem can be considered for different visibility models namely limited visibility, i.e., a robot can observe the plane up to a fixed radius around it and obstructed visibility, i.e., the robots block each other's vision.

**Author Contributions:** Conceptualization, All authors; methodology, All authors; software, All authors; validation, All authors; formal analysis, All authors; investigation, All authors; resources, All authors; data curation, All authors; writing—original draft preparation, All authors; writing—review and editing, All authors; visualization, All authors; supervision, K.M.; project administration, All authors; funding acquisition, All authors. All authors have read and agreed to the published version of the manuscript.

**Funding:** This research received no external funding.

**Informed Consent Statement:** Not applicable.

**Data Availability Statement:** Not applicable.

**Conflicts of Interest:** The authors declare no conflict of interest.

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
