# Peer review of "k-Circle Formation and k-epf by Asynchronous Robots"

_algorithms, doi:10.3390/a14020062_

Round 1

Reviewer 1 Report

Summary of contributions:
-------------------------------------
This paper introduces and studies the k-circle formation problem. This problem concerns a team of n mobile entities moving in the Euclidean, where each entity can observe a set of m "fixed points" in the plane. The task for the entities is to form m circles, each around one of m "fixed points", with the following constraints: (1) n=km, (2) each circle is centered at each of the fixed points and is formed by k entities located in distinct positions, and (3) the circles formed by the robots must be disjoint (no robot can be located on any two distinct circles).

According to the current literature, mobile entities are computationally weak:
they are silent, autonomous, anonymous, oblivious, disoriented, and homogeneous. On the contrary, they have unlimited visibility and can observe all the plane. They operate in the classical Look-Compute-Move model and according to the Asynchronous (ASYNC) scheduler: entities do not have a common notion of time.

Authors provide interesting motivations for the introduction of this problem, both theoretical and practical. From the theoretical point of view, the k-circle formation is a hybrid problem in between the partitioning problem and the circle formation problem.

Authors claim to provide three main contributions:
(1) characterization of all the initial configurations and the values of k for which the k-circle formation problem is deterministically unsolvable in the considered scenario;
(2) a deterministic distributed algorithm able to solve the k-circle formation problem for the rest of the initial configurations.
BUT this algorithm works under additional assumptions: the entities have an agreement on the direction and orientation of the y-axis.
(3) a formal statement proving that the k-circle formation problem is deterministically solvable if and only if the k-epf problem is deterministically solvable for the initial configurations with distinct robot positions. The k-epf problem is defined as follows: given m > 0 fixed points and n = km robots for some positive integer k, the k-epf problem asks exactly k robots to reach and remain in each fixed point.

General comments:
-------------------------------------
The paper is well written and well organized. The studied problem is interesting and deserves to be investigated. Concerning the obtained results, the first two appear to be correct, even if the additional assumptions concerning the agreement on the direction and orientation of the y-axis diminish its reach. In fact, the problem should be studied when the entities are completely disoriented. This, on the one hand, would bring out many more unsolvable configurations, while on the other it would make the algorithm more complex due to further symmetries that would lead to considering moves with multiple entities active at the same time.

Concerning the last result, it is provided by Theorem 3. This theorem formally states that "For a given initial configuration with distinct robot positions and a positive integer k, the k-circle formation problem is deterministically solvable if and only if the k-epf problem is deterministically solvable".

It is the opinion of the referee that to prove the statement, authors use an approach which fails from a methodological point of view.

For instance, to prove the "necessity" part, they consider *any* algorithm A1 able to solves the k-circle formation problem ad *modify it* into an algorithm A1' so that A1' unsure that a certain Property 1 holds. This property is necessary to correctly later activate an algorithm A2 that bring the configuration fulfilling the property to a final configuration solving the k-epf problem. Hence, they prove this art of the statement by claiming that A1' composed with A2 form an algorithm A able to solve k-epf.

This approach fails because the authors should formally prove that the modification made to A1 to get A1' does not affect the correctness of A1; only with this proof the authors can assume the correctness of A1'. Since A1 is taken as a black box, it is impossible to bu sure that A1' is correct!
The same approach is used for the "sufficiency" part of the proof. They take any algorithm E able to solve the k-epf problem and modify it to solve the k-circle formation.

Final comment: a revision is required
------------------------------------
The authors should modify Section 6. They can still prove that k-circle formation --> k-epf formation. This can be done by using the proposed algorithm. Authors could provide evidence that by modifying the proposed algorithm, a solution for the k-epf can be derived.
Conversely, they cannot formally state k-epf formation --> k-circle formation.

If the proposed change were made, the resulting document would be worth publishing.

Minor comments:
-------------------------------------
- you should emphasize that you refer to the "circle formation" and not to the "uniform circle formation" (eg, see reference 20), which is much harder to solve.
- l 198: you should observe that, according to the agreement on the direction and orientation of the y-axis, other possible symmetries (eg, rotational symmetries) do not apply

Author Response

Response to Reviewer 1 Comments

------------------------------------------------------------------------------------------------------------------------

Comment 1: A revision is required.

The authors should modify Section 6. They can still prove that k-circle formation --> k-epf formation. This can be done by using the proposed algorithm. Authors could provide evidence that by modifying the proposed algorithm, a solution for the k-epf can be derived.
Conversely, they cannot formally state k-epf formation --> k-circle formation.

Response 1.

We are grateful for the reviewer’s comment. We agree that the proof for Theorem 3 given in the version submitted on 31.12.2020, fails from a methodological point of view. As suggested by the reviewer, we have modified the section 6 (Lines-733-799). We have modified the statement of Theorem 3 (Lines-737-738). In Theorem 3, we have formally stated that “For a given initial configuration with distinct robot positions and a given positive integer k, if the k-circle formation problem is deterministically solvable then the k-epf problem is also solvable.”

In order to prove Theorem 3, as suggested by the reviewer, we have modified the proposed algorithm $AlgorithmOneAxis$, such that the modified algorithm solves the k-epf problem (Lines-742-793). The modified algorithm is $Algokepf$, described in subsection 6.1 (Lines-742-793). The correctness of $Algokepf$ follows from Theorem 2 (Line no-692-694), Lemma 10 (Lines-761-781) and Theorem 4 (Lines-782-793).

As pointed by the reviewer we could not formally state that the k-epf formation implies the k-circle formation. We have removed this part.

Comment 2.

You should emphasize that you refer to the "circle formation" and not to the "uniform circle formation" (eg, see reference 20), which is much harder to solve.

Response 2.

We agree to this comment. We have added the line “Note that the circles do not require to be necessarily uniform. In other words the k robots forming one particular circle have identical

distance from a particular fixed point, but their mutual distances can be anything.” in Section 1 (Lines-38-40).

Comment 3.

l 198: you should observe that, according to the agreement on the direction and orientation of the y-axis, other possible symmetries (eg, rotational symmetries) do not apply.

Response 3.

We agree to the reviewer and we have added the observations “Since the robots have an agreement on the direction and orientation of the y-axis, the configuration can not admit translational symmetry or rotational symmetry.”(Lines-208-210).

Reviewer 2 Report

Comments:

This manuscript had investigated the k-circle formation problem and the k-epf problem for a set of autonomous, asynchronous robots. A deterministic distributed algorithm has been designed to solve the k-circle formation problem in the setting where the robots have an agreement on the common direction and orientation of one of the axes. This paper is well-organized and provides some interesting results in some sense. However, the following comments should still be taken into account.

  1. The introduction section provides some literature review on the swarm robotics. However, more recent researches should be presented so that the paper can be more persuasive. Besides, the main contributions of the paper need to be further summarized.

  1. The organization of the introduction should be more logical and a more concise discussing bringing the considered problem of the paper should better be presented.

  1. The writing quality and English expressions can still be improved. The reviewer can still find out some mistakes in this version. For example,in the page 11 of 29, “Since r_i is a candidate robot \bar{r_i(t)q} (excluding point q) does not contain any robot positions” should be “Since r_i is a candidate robot \bar{r_i(t)q} (excluding point q) not containing any robot positions”. Furthermore, the other parts of the manuscript should be double-checked and corresponding revisions should be made.

  1. What are the difficulties in the practical application of the proposed control  algorithm and what solutions do the author adopt to deal with them?

  1. In my opinion, the expression of the conclusion part is almost the same as the abstract part. Thus, it may be better if the authors can further revise and summary the most significant results of the paper.

In conclusion, this paper had given some interesting results. I would like to suggest that this paper can be accepted for publication after some revisions.

Author Response

Response to Reviewer 2 Comments

------------------------------------------------------------------------------------------------------------------------

Comment 1.

The introduction section provides some literature review on the swarm robotics. However, more recent researches should be presented so that the paper can be more persuasive. Besides, the main contributions of the paper need to be further summarized.

Response 1.

We are grateful to the reviewer’s valuable comments. We agree with the reviewer. We have added the following recent researches :

1. Arbitrary pattern formation by asynchronous opaque robots with lights. Theoretical Computer Science 2021. (Lines-108-112) Reference-25.

2. Arbitrary pattern formation on infinite grid by asynchronous oblivious robots. Theoretical Computer Science 2020. (Lines-112-113) Reference-26.

3. Uniform Circle Formation for Swarms of Opaque Robots with Lights. Stabilization, Safety, and Security of Distributed Systems 2018 (Lines-122-123) Reference-28.

4. Gathering over Meeting Nodes in Infinite Grid. Conference on Algorithms and Discrete Applied Mathematics. Springer, 2020 (Line no-132) Reference-23.

As suggested by the reviewer we have further summarized the main contributions in Our Contribiutions (Lines-139-150).

Comment 2.

The organization of the introduction should be more logical and a more concise discussing bringing the considered problem of the paper should better be presented.

Response 2.

We have added a discussion (Lines-28-33) before bringing the considered problem of the paper. In this paragraph, we have discussed the problems in this area of research work which considers pre-fixed points in the plane. We have discussed the circle formation problem later in lines-80-83.

Comment 3.

The writing quality and English expressions can still be improved. The reviewer can still find out some mistakes in this version. For example,in the page 11 of 29, “Since r_i is a candidate robot \bar{r_i(t)q} (excluding point q) does not contain any robot positions” should be “Since r_i is a candidate robot \bar{r_i(t)q} (excluding point q) not containing any robot positions”. Furthermore, the other parts of the manuscript should be double-checked and corresponding revisions should be made.

Response 3.

We have re-written the statement as “Since r_i is a candidate robot, the line segment \bar{r_i ( t ) q } (excluding point q) can not possibly contain any robot positions other than r_i ( t ) .”(Lines-393-394). We have re-written all the statements in this subsection 4.4 (Lines-381-382, 387-388, 393-394, 401-402, 405-406). We have checked the document and corrected the mistakes we have found, as suggested by the reviewer.

Comment 4.

What are the difficulties in the practical application of the proposed control  algorithm and what solutions do the author adopt to deal with them?

Response 4.

The biggest challenge in the practical application of swarm robotics would be the assumption of infinite precision. In this paper, we have assumed that a robot always moves in a straight line. However, in the physical world, a robot can not move in a straight line without any error. Also, the assumption that a robot can move infinitesimal distance is not possible in the physical world. With the current model of swarm robotics we do not have a solution to overcome these problems. We hope that with further developments in this research area, the model would be competent for practical applications in the future.

Comment 5.

In my opinion, the expression of the conclusion part is almost the same as the abstract part. Thus, it may be better if the authors can further revise and summary the most significant results of the paper.

Response 5.

We have modified the conclusion part. We have added the most significant results of the paper, as suggested by the reviewer (Lines-801-824).

Reviewer 3 Report

Authors study the k-circle formation problem in the Euclidean plane by oblivious anonymous robots in the asynchronous setting. There are a fixed set of m points given and the total number of robots is n=mk, meaning that in each of the m points as the center of the circle, there are exactly k robots each. They studied the problem under one axis agreement. The first provided an impossibility of solving the problem starting from some initial configurations of robots with some combinations of n and k parameters. Except the impossibility configurations, they have a deterministic algorithm solving the problem. The algorithm seems correct and analysis is detailed. The techniques seem non-trivial. 

The motivation is clear and the authors provided a connection to this problem to partitioning and n-circle formation (m=1) problems. They also showed provided a connection between k-circle and k-epf problem, showing that k-circle formation is deterministically solvable if and only if k-epf is deterministically solvable. These results are interesting and I recommend acceptance. 

One comment for minor revision: What is the running time of these algorithms. Besides correctness, runtime is also a crucial issue and it will be interesting to have an analysis of runtime. At least some discussion regarding that.  

Author Response

Response to Reviewer 3 Comments

------------------------------------------------------------------------------------------------------------------------

Comment 1.

One comment for minor revision: What is the running time of these algorithms. Besides correctness, runtime is also a crucial issue and it will be interesting to have an analysis of runtime. At least some discussion regarding that.  

Response 1.

We are grateful for the reviewer’s valuable comment. We have discussed the running time of $AlgorithmOneAxis$ with the notion of an epoch. An epoch is the time interval in which all the robots in the configuration have performed their LCM cycles at least once. We have assumed that the robots have rigid motion. In this setting, we have given a loose upper bound for the running time of $AlgorithmOneAxis$ (Lines-712-731).